# Learning Dynamics of RNNs in Closed-Loop Environments

**Yoav Ger**
yoav.ger@campus.technion.ac.il

**Omri Barak**
omri.barak@gmail.com

Rappaport Faculty of Medicine and Network Biology Research Laboratory
Technion, Israel Institute of Technology

## Abstract

Recurrent neural networks (RNNs) trained on neuroscience-inspired tasks offer powerful models of brain computation. However, typical training paradigms rely on open-loop, supervised settings, whereas real-world learning unfolds in closed-loop environments. Here, we develop a mathematical theory describing the learning dynamics of linear RNNs trained in closed-loop contexts. We first demonstrate that two otherwise identical RNNs, trained in either closed- or open-loop modes, follow markedly different learning trajectories. To probe this divergence, we analytically characterize the closed-loop case, revealing distinct stages aligned with the evolution of the training loss. Specifically, we show that the learning dynamics of closed-loop RNNs, in contrast to open-loop ones, are governed by an interplay between two competing objectives: short-term policy improvement and long-term stability of the agent-environment interaction. Finally, we apply our framework to a realistic motor control task, highlighting its broader applicability. Taken together, our results underscore the importance of modeling closed-loop dynamics in a biologically plausible setting.

## 1 Introduction

Recurrent neural networks (RNNs) are widely used to study dynamic processes in both neuroscience and machine learning [1, 2, 3, 4, 5]. Their recurrent architecture, reminiscent of brain circuits, enables them to form internal representations such as memories of past inputs [6, 7]. When coupled with an environment, these memories can support the formation of internal world models that help agents achieve their goals. This capability is critical in real-world settings, which are often ambiguous and only partially observable. However, the conditions under which these models emerge during training remain poorly understood, particularly in closed-loop agent–environment interactions, where outputs influence subsequent inputs.

Despite the ubiquity of closed-loop interaction in biological learning, most RNN training to date has relied on supervised learning in an open-loop setting, with theoretical work primarily focused on characterizing the properties of the resulting solutions [8, 9, 10, 11, 12, 13, 14, 15]. Motivated by recent analytical understanding of feedforward networks [16], several studies have begun to investigate the learning dynamics themselves, though they remain confined to simplified open-loop regime [17, 18, 19, 20, 21, 22]. These approaches typically assume *i.i.d.* inputs that are independent of the network's outputs, thereby omitting the feedback-driven environments in which biological agents learn. As a result, the learning dynamics that arise in closed-loop settings remain largely unexplored. In particular, it is unclear whether such conditions induce distinct learning trajectories or solution classes compared to conventional open-loop training.

Training agents in environments is typically approached through reinforcement learning (RL), encompassing methods such as policy gradients and actor-critic algorithms [23, 24, 25]. While RL has

39th Conference on Neural Information Processing Systems (NeurIPS 2025).

achieved remarkable success in complex tasks, the complexity of modern architectures and training pipelines often obscures theoretical understanding of the learning dynamics [26, 27, 28]. Here, we focus on a simple, tractable setting where a single recurrent network is trained with policy gradients on tasks that preserve key features of agent–environment learning. These include feedback that induces correlated inputs and partial observability, requiring internal representations.

Using this setting, we develop a mathematical framework for learning in closed-loop environments. We begin by showing that closed-loop and open-loop training produce fundamentally different learning dynamics, even when using identical architectures and converging to the same final solution. To investigate this divergence, we focus on the largely understudied dynamics of closed-loop RNNs. Specifically, we show that tracking the eigenvalues of the coupled agent–environment system (rather than the RNN alone) is both necessary and sufficient to uncover the structure of the learning process, which unfolds in distinct stages reflected in the spectrum and training loss. Notably, closed-loop learning gives rise to a natural trade-off between two competing objectives: myopic, short-sighted policy improvement and long-term system-level stability. Finally, we demonstrate that similar learning dynamics arise in a more complex motor control task, with RNNs progressing through stages similar to those observed in human experiments.

## 2  Preliminaries

We begin by introducing our framework, comprising the environment (a double integrator task) and the agent (an RNN), which is trained under either closed-loop or open-loop conditions. Throughout, we use bold lowercase letters for vectors (e.g., $z$), bold uppercase letters for matrices (e.g., $W$), and non-bold letters for scalars. The identity matrix is denoted by $I$. We define the scalar overlap between two vectors $p$ and $q$ as $\sigma_{pq} = p^\top q$. The spectral radius of a matrix is denoted by $\rho(\cdot)$, and we write $\chi_{(\cdot)}(\lambda)$ for the characteristic polynomial of a matrix with eigenvalues $\lambda$.

**Double integrator task**   Our task environment is the classic discrete-time double integrator control problem [29, 30], described by the following linear state-space model:

$$x_{t+1} = A x_t + B\, u_t, \quad \text{where} \quad A = \begin{bmatrix} 1 & 1 \\ 0 & 1 \end{bmatrix}, \quad B = \begin{bmatrix} 0 \\ 1 \end{bmatrix} \tag{1}$$

Here, $x_t = (x_t^{(1)}, x_t^{(2)})^\top \in \mathbb{R}^2$ represents the position and velocity of a unit mass, while $u_t \in \mathbb{R}$ is the control signal applied at each time step to steer the mass toward the target state $x^* = (0, 0)^\top$. Note that acceleration is precisely equal to the control input (i.e., $\ddot{x} = u_t$). Since we are interested in *partial observability*, our observation matrix satisfies $\text{rank}(C) < \text{rank}(A)$. Specifically, only the position is measured,

$$y_t = C\, x_t, \quad C = \begin{bmatrix} 1 & 0 \end{bmatrix} \tag{2}$$

so that $y_t \equiv x_t^{(1)}$. The goal is to generate a sequence of control signals that minimizes the average squared Euclidean distance to the target state without expending excessive force,

$$\min_{u_{1:T}} \frac{1}{T} \sum_{t=1}^{T} \|x_t - x^*\|^2 + \beta \sum_{t=1}^{T} u_t^2 \tag{3}$$

where $\beta$ balances the cost of state error against control effort.

**RNN model**   We model our RNN-based agent as a large network of $N$ recurrently connected neurons that interact via continuous firing rates. We adopt a discrete-time RNN to align with the discrete nature of the control task. The network state dynamics and output are given by:

$$h_{t+1} = \phi\big(W\, h_t + m\, y_{t+1}\big), \quad u_{t+1} = z^\top h_{t+1} \tag{4}$$

where $h_t \in \mathbb{R}^N$ is the hidden state of the RNN, $\phi(\cdot)$ denotes the activation function, $W \in \mathbb{R}^{N \times N}$ is the recurrent weight matrix, and $m, z \in \mathbb{R}^{N \times 1}$ are the input and output weight vectors, respectively. Here, $y_{t+1}$ is the mass position and $u_{t+1}$ the control signal. We also validate our findings in a continuous-time RNN model (Appendix A.1).

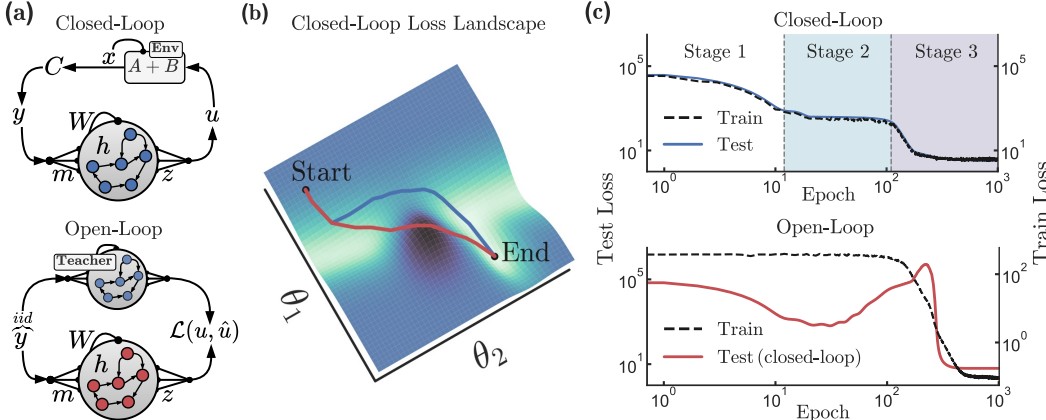

Figure 1: **(a)** Schematic of learning setups: In the closed-loop setting (top), the RNN output $u$ is fed into the environment, which evolves according to the system dynamics and produces the next input $y$ to the network. In contrast, the open-loop setting (bottom) lacks feedback: a student RNN is trained via supervised learning to imitate a pre-trained teacher mapping *i.i.d.* inputs $y$ to target outputs $u$. **(b)** Conceptual illustration of our hypothesis: optimization trajectories for closed-loop (blue) and open-loop (red) RNNs explore different regions of the closed-loop loss landscape (darker colors indicate higher loss). **(c)** Summary of results: the closed-loop RNN (top) progresses through three distinct learning stages, while the open-loop RNN (bottom) shows a sharp test loss peak. Note that the open-loop train loss (dashed black) drops by roughly one order of magnitude while the closed-loop test loss (red solid) drops by four, highlighting its limited sensitivity to the environment. Both panels use log scale; dashed and solid lines show train and test loss on right and left axes, respectively.

**Closed-loop learning setup**   In this setup, the RNN agent learns through repeated interaction with the environment. At each time step, the environment's state $x$ evolves according to Eq. 1, and a projection of this state, $y$, is fed as input to the RNN. From the agent's internal state, $h$, an output $u$ is produced, which serves as a control signal for the environment, thus closing the feedback loop (Fig. 1a; top). The initial mass state is sampled uniformly: $x_0 \sim \mathcal{U}([-2, 2]^2)$. The RNN is trained via policy gradients [31, 32] to minimize the objective in Eq. 3. This process can be interpreted as a form of adaptive control [33], where policy learning and system identification occur simultaneously (see Appendix B). While classical control often includes a control penalty, we adopt an unregularized version [34] by setting $\beta = 0$. This simplification does not affect our main findings, though it may influence the final solution the RNN settles on (see Stage 3 and Appendix A.2).

**Open-loop learning setup**   Following the standard teacher–student framework [35, 36], we formulate an open-loop supervised setup in which a student RNN does not interact with the environment but is instead trained to imitate an input–output mapping defined by a teacher network pre-trained on the task (Fig. 1a; bottom). To ensure *i.i.d.* training data, we inject white noise inputs into the teacher and use its outputs as targets. Formally, the training objective is given by $\min \frac{1}{T} \sum_{t=1}^{T} (\hat{u}_t - u_t)^2$, where $\hat{u}_t$ and $u_t$ denote the student's prediction and the teacher's target, respectively. Note that the double-integrator is a special control task that admits an analytically optimal solution (see LQR in Appendix B). Accordingly, we also supplemented our results using the corresponding LQR controller as a teacher (Appendix A.5).

**Training details and RNN initialization**   Both the closed-loop and open-loop RNNs share the same architecture and are initialized identically. Each network consists of $N = 100$ neurons, an episode length of $T = 50$, and the hyperbolic tangent activation function, $\phi(\cdot) = \tanh$. The input and output weight vectors were initialized independently from $\mathcal{N}(0, 1/N)$, and the recurrent weight matrix $W$ was initialized from $\mathcal{N}(0, g^2/N)$, where $g$ controls the initial recurrent strength. Training was performed using stochastic gradient descent on $W$, $m$, and $z$, with a learning rate of $\eta = 10^{-2}$, a batch size of 100, and gradient clipping (2-norm capped at 1) to avoid exploding gradients.

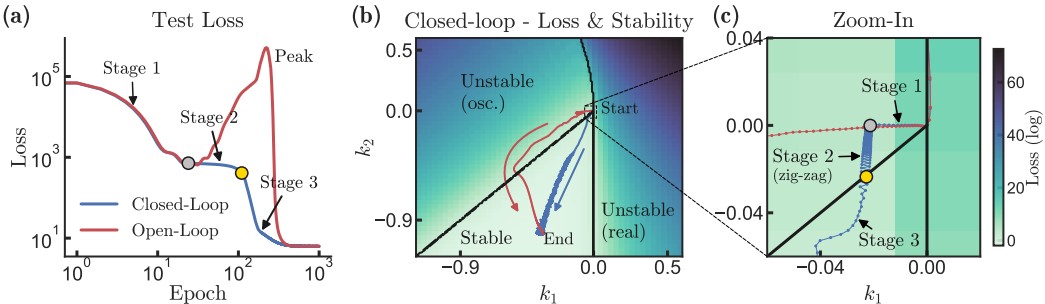

Figure 2: **(a)** Closed-loop test loss (reproduced from Fig. 1c). Despite identical architectures, the two RNNs exhibit distinct learning dynamics under closed-loop and open-loop training. The closed-loop RNN progresses through three distinct learning stages, whereas the open-loop RNN displays a sharp loss peak. Gray and yellow markers indicate the end of Stages 1 and 2, respectively, for the closed-loop RNN. **(b)** Numerical estimation of the effective RNN gain $\boldsymbol{K}_{\text{eff}}$ during training, projected onto the $(k_1, k_2)$ plane (closed-loop shown in blue; open-loop in red). Although both RNNs begin and end in similar regions, they follow markedly different trajectories across the loss landscape. The background color represents the logarithm of the loss; darker regions correspond to higher loss. The solid black curve separates three distinct stability regimes. **(c)** Zoomed-in view of panel (b), highlighting the divergence point between the two trajectories (gray marker). Gray and yellow markers match those in panel (a). Notably, the end of Stage 2 for the closed-loop RNN (yellow marker) coincides with a stability phase transition in the $(k_1, k_2)$ plane.

## 3 Closed-loop and open-loop RNNs exhibit different learning dynamics

We begin with a simple question: if all else is equal, will two identical RNNs—one trained in a closed-loop setup and the other in an open-loop setup—exhibit the same learning trajectories? We hypothesize they will not, as the open-loop agent lacks feedback from the environment and is therefore blind to the consequences of its actions. As a result, producing actions that closely match those of the teacher (and thus reduce loss in the open-loop sense) may be detrimental once the environment is introduced (Fig. 1b). To test this, we train two identical nonlinear RNNs (initialized with $g = 0.1$) under the two paradigms and find that their learning dynamics diverge substantially (Fig. 1c). While both RNNs show similar initial improvement, the closed-loop RNN quickly enters a plateau phase. In contrast, its open-loop counterpart exhibits a sharp peak in closed-loop test loss, indicating a breakdown in performance. Although the loss eventually recovers, it is accompanied by only a minor improvement in open-loop training loss, underscoring the setup's insensitivity to control consequences (Fig. 1c; bottom). We further verified that this divergence between closed- and open-loop learning is not specific to the double-integrator task, but persists across a broader class of control problems that generalize it (Appendix A.6).

To formalize this divergence concretely, we build on concepts from control theory. Specifically, we consider the closed-loop stability of the double integrator system under linear state feedback [37], defined by:

$$\boldsymbol{M}_{\text{cl}} = \boldsymbol{A} - \boldsymbol{B}\boldsymbol{K} = \begin{bmatrix} 1 & 1 \\ 0 & 1 \end{bmatrix} - \begin{bmatrix} 0 \\ 1 \end{bmatrix} \begin{bmatrix} k_1 & k_2 \end{bmatrix} = \begin{bmatrix} 1 & 1 \\ -k_1 & 1 - k_2 \end{bmatrix} \tag{5}$$

Sweeping over a range of feedback gains $(k_1, k_2)$, we simulate system trajectories and compute the total squared distance from the target state. Additionally, by analyzing the eigenvalues of $\boldsymbol{M}_{\text{cl}}$, we identify regions in parameter space corresponding to stable and unstable dynamics (Fig. 2b and Appendix B.4). Next, to embed the high-dimensional nonlinear RNN policy into this interpretable 2D space, we sample trajectories from both RNNs during training, recording the full system state $\boldsymbol{x}_t$ and the corresponding control output $u_t$ (noting that the RNN observes only the position of the mass). We then fit a linear model of the form:

$$u_t \approx -k_1 x_t^{(1)} - k_2 x_t^{(2)} \tag{6}$$

yielding an empirical estimate, which we denote as the *effective* feedback gain, $\boldsymbol{K}_{\text{eff}} = [k_1 \ k_2]$. Note that this is only an approximation. An exact analytical map to $(k_1, k_2)$ is possible only when the RNN is linear and the dynamics lie in an effectively two-dimensional subspace (Appendix C.9).

As shown in Fig. 2b, and consistent with our hypothesis, the two RNNs follow distinct trajectories within this low-dimensional subspace defined by the closed-loop system. Closer inspection of early training (Fig. 2c) reveals three notable observations. First, the divergence between open- and closed-loop learning emerges early, at the end of Stage 1 (Fig. 2c gray marker). Second, the progression from Stage 1 to Stage 2 of the closed-loop RNN unfolds in a zig-zag trajectory, reflecting non-monotonic updates in the effective feedback gain. Third, the transition from Stage 2 to Stage 3 in the closed-loop RNN (when the plateau ends) coincides with a shift from unstable to stable dynamics in the $(k_1, k_2)$ plane (Fig. 2c yellow marker).

Before shifting our focus to the closed-loop RNN, we note that the dynamics of learning in the open-loop setup is not trivial itself. If the network were linear, the open-loop loss would be equivalent to learning a linear filter via gradient descent. This setting was recently analyzed analytically [18], and we reproduce similar results in Appendix A.4 for completeness. Next, to better understand the three observations outlined above, we focus on the less explored case of closed-loop learning.

## 4 Analytic results of closed-loop learning dynamics

Having shown that closed-loop RNNs exhibit distinct learning dynamics from their open-loop counterparts, we now turn to our analytical treatment, beginning with three simplifying assumptions. First, we replace the nonlinear activation with a linear one, $\phi(\cdot) = \mathrm{id}$. Second, consistent with prior work [17, 38], we observe that training induces low-rank connectivity, often rank-1 (Appendix A.3), and thus we set $\boldsymbol{W} = \boldsymbol{u}\,\boldsymbol{v}^\top$ with $\boldsymbol{u}, \boldsymbol{v} \in \mathbb{R}^{N \times 1}$. Third, the input weights $\boldsymbol{m}$ are randomly set and fixed during training. These assumptions simplify analysis without sacrificing generality.

### 4.1 Coupled closed-loop system derivation

To analyze the interaction of the closed-loop RNN, we adopt a unified framework, standard in control theory, that combines the dynamics of both the RNN and the environment. Since both are linear, we stack the environment state $\boldsymbol{x}_t$ and the RNN hidden state $\boldsymbol{h}_t$ into a joint state vector $\boldsymbol{s}_t = (\boldsymbol{x}_t, \boldsymbol{h}_t)^\top$, yielding the coupled linear system:

$$
\boldsymbol{s}_{t+1} = \boldsymbol{P}\,\boldsymbol{s}_t, \quad \text{where} \quad \boldsymbol{P} = \begin{bmatrix} \overbrace{\boldsymbol{A}}^{\text{env. dynamics}} & \overbrace{\boldsymbol{B}\,\boldsymbol{z}^\top}^{\text{control output}} \\ \underbrace{\boldsymbol{m}\,\boldsymbol{C}\,\boldsymbol{A}}_{\text{input feedback}} & \underbrace{\boldsymbol{W}}_{\text{RNN dynamics}} \end{bmatrix} \tag{7}
$$

The matrix $\boldsymbol{P} \in \mathbb{R}^{(2+N) \times (2+N)}$ defines the full closed-loop dynamics, with stability governed by its eigenvalues. For a general $\boldsymbol{W}$, these are found by solving a self-consistent characteristic equation (Appendix C.2). In the specific case of rank-1 connectivity (i.e., $\boldsymbol{W} = \boldsymbol{u}\boldsymbol{v}^\top$), the hidden state $\boldsymbol{h}_t$ remains confined to the subspace spanned by $\boldsymbol{m}$ and $\boldsymbol{u}$, and can be expressed as $\boldsymbol{h}_t = \kappa_t^{(m)}\,\boldsymbol{m} + \kappa_t^{(u)}\,\boldsymbol{u}$. This allows us to formulate a reduced four-dimensional system:

$$
\boldsymbol{s}_t^{(\text{eff})} = \begin{bmatrix} x_t^{(1)} \\ x_t^{(2)} \\ \kappa_t^{(m)} \\ \kappa_t^{(u)} \end{bmatrix} \in \mathbb{R}^4, \qquad \boldsymbol{s}_{t+1}^{(\text{eff})} = \boldsymbol{P}_{\text{eff}}\,\boldsymbol{s}_t^{(\text{eff})}, \quad \text{where} \quad \boldsymbol{P}_{\text{eff}} = \begin{bmatrix} 1 & 1 & 0 & 0 \\ 0 & 1 & \sigma_{\boldsymbol{zm}} & \sigma_{\boldsymbol{zu}} \\ 1 & 1 & 0 & 0 \\ 0 & 0 & \sigma_{\boldsymbol{vm}} & \sigma_{\boldsymbol{vu}} \end{bmatrix} \tag{8}
$$

This *effective* system is governed by just four scalar order parameters–overlaps denoted by $\sigma$–with eigenvalues satisfying:

$$
\chi_{\boldsymbol{P}}(\lambda) = \lambda^3 + (-\sigma_{\boldsymbol{vu}} - 2)\,\lambda^2 + (2\sigma_{\boldsymbol{vu}} - \sigma_{\boldsymbol{zm}} + 1)\,\lambda + (\sigma_{\boldsymbol{vu}}\sigma_{\boldsymbol{zm}} - \sigma_{\boldsymbol{vu}} - \sigma_{\boldsymbol{zu}}\sigma_{\boldsymbol{vm}}) \tag{9}
$$

While $\boldsymbol{P}$ (with $\boldsymbol{W} = \boldsymbol{u}\boldsymbol{v}^\top$) and $\boldsymbol{P}_{\text{eff}}$ produce identical within-episode trajectories due to spectral equivalence, their learning dynamics are not similar: gradients applied to the order parameters in $\boldsymbol{P}_{\text{eff}}$ do not directly correspond to weight updates in the full RNN. Nonetheless, we numerically find that learning in the reduced system captures key features of high-dimensional RNN learning, as we show next, stage by stage. In Appendix C we provide a full derivation of both systems and demonstrate their spectral identity (within-episode).

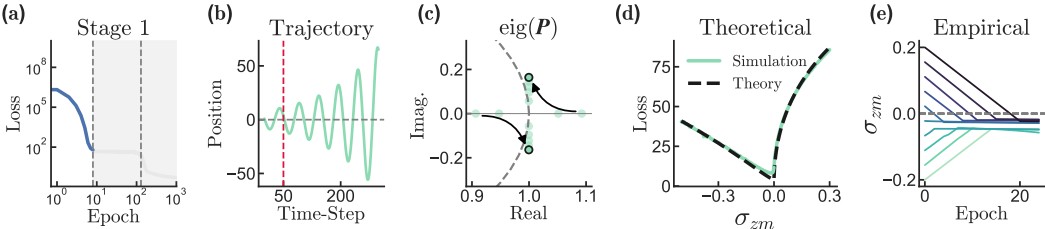

Figure 3: Stage 1 - Negative position policy. **(a)** Training loss rapidly decreases during Stage 1. **(b)** Example trajectory of mass position under RNN control at the end of Stage 1, exhibiting oscillatory divergence; note, the red dashed line indicates the episode length used for training. **(c)** This oscillatory instability is explained by inspecting the eigenspectrum of $P$, which reveals the emergence of a dominant complex-conjugate pair during training with $\rho(P) > 1$. Arrow indicates trajectory; darker marker denotes final position. **(d)** The effective loss (dashed), predicted by the order parameter $\sigma_{zm}$ alone, shows strong agreement with simulation (solid). **(e)** As predicted by the asymptotic loss landscape in (d), despite initializing many RNNs with different initial overlaps, all networks converge to a small and negative $\sigma_{zm}$ by the end of Stage 1.

## 4.2 Three stages of learning: control, representation, and refinement

**Stage 1 - Negative position policy**    In the first stage, the loss rapidly decreases (Fig. 3a) as the RNN adopts a "negative-position" policy, $u_t \propto -\text{position}$ (Fig. 3b). This behavior resembles a proportional (P) controller [37], where the control signal is directly tied to the instantaneous position error. While this provides immediate corrective feedback, it lacks any internal representation of the mass's velocity (i.e., no derivative or integral components as in PID controllers). Consequently, this policy induces instability in the combined system $P$, as evidenced by an eigenvalue with magnitude greater than one (Fig. 3c), which dominates the loss during this stage.

More specifically, since gradient descent induces only minimal change to $W = uv^\top$ during Stage 1, we set it to zero, which simplifies Eq. 9 to take a quadratic form:

$$\chi_P(\lambda) = \lambda^2 - 2\lambda + (1 - \sigma_{zm}) \tag{10}$$

with eigenvalues given by $\lambda = 1 \pm \sqrt{\sigma_{zm}}$, where $\sigma_{zm}$ quantifies the alignment between the input and output vectors. This corresponds to two types of instability, as illustrated in Fig. 2b: a real dominant eigenvalue when $\sigma_{zm} > 0$, and a complex-conjugate pair when $\sigma_{zm} < 0$. This analysis enables a closed-form approximation of the loss in Stage 1 as a function of $\sigma_{zm}$ alone, which shows excellent agreement with the empirical loss (Fig. 3d).

In addition, the analysis reveals an asymmetric loss landscape around $\sigma_{zm} = 0$, with a sharper rise for positive overlaps compared to negative ones. This asymmetry drives learning toward small negative values of $\sigma_{zm}$ during this stage (Fig. 3e). Taking the limit $T \to \infty$ gives $\sigma_{zm} \to 0$. For finite $T$, however, the asymmetric shape of the loss pushes $\sigma_{zm}$ to small negative values. This can also be understood as the need to construct early-episode *control* at the expense of long-term behavior, as we expand next. A full derivation of Stage 1 appears in Appendix C.5.

**Stage 2 - Building a world model**    In Stage 2, the loss enters a plateau phase (Fig. 4a), which ends when the RNN learns to steer the mass to the target via a highly underdamped response with sustained oscillations (Fig. 4b). This phase concludes when the closed-loop dynamics stabilize, as the dominant eigenvalues of $P$ enter the unit disk (Fig. 4c). Reaching this stable regime requires the RNN to infer velocity, a hidden variable not directly available from the input. To understand how stability is achieved, we examine how gradient descent updates alter the spectral structure of $P$.

Unlike Stage 1, the recurrent weights, $W = uv^\top$, now evolve and must be included in the spectral analysis. From Stage 1, we know that the characteristic polynomial admits a complex-conjugate pair of roots, implying that the third root is real. Applying Vieta's formula to $\chi_P(\lambda)$ yields:

$$|\lambda_1|^2 = (2\sigma_{vu} - \sigma_{zm} + 1) + (-\sigma_{vu} - 2)\lambda_3 + \lambda_3^2 \tag{11}$$

where $\lambda_1$ and $\lambda_3$ denote the complex and real roots of Eq. 9, respectively. To approximate $\lambda_3$, we apply first-order perturbation theory around the non-zero eigenvalue $\sigma_{vu}$ of the rank-one connectivity

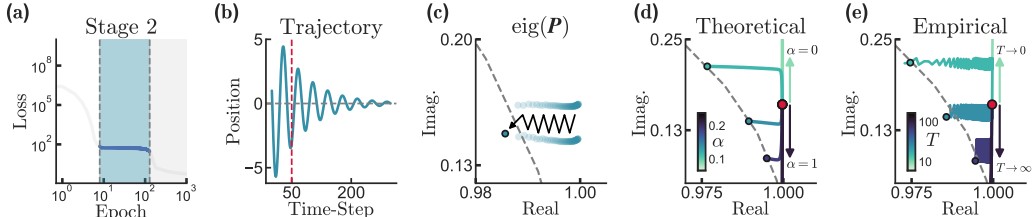

Figure 4: Stage 2 - Building a world model. **(a)** Training loss enters a plateau phase. **(b)** Example trajectory of mass position under RNN control at the end of Stage 2, showing an underdamped response with sustained oscillations that eventually converge to the target (note the time-step on the x-axis). **(c)** Empirical evolution of one of the dominant complex-conjugate eigenvalues during Stage 2, exhibiting a zig-zag trajectory inward toward the unit disk (dashed). Arrow indicates trajectory; darker marker denotes final position. **(d)** Theoretical trajectories of the dominant eigenvalue optimizing the surrogate loss across different $\alpha$ values: optimizing only short-term loss ($\alpha = 0$) drives the eigenvalue upward along the imaginary axis; optimizing only long-term loss ($\alpha = 1$) causes direct descent; intermediate values ($0 < \alpha < 1$) yield trajectories consistent with empirical behavior; red marker indicates the initial condition. **(e)** Empirical validation: initializing the RNN from the same starting point (red dot) and varying episode length $T$ produces trajectories that qualitatively match the theoretical surrogate predictions.

matrix, yielding:

$$\lambda_3 \approx \sigma_{vu} + \frac{\sigma_{zu}\,\sigma_{vm}}{(\sigma_{vu})^2 - 2\sigma_{vu} - \sigma_{zm} + 1} + \mathcal{O}(\epsilon^2) \tag{12}$$

Substituting Eq. 12 into Eq. 11 yields a closed-form approximation for the dominant eigenvalue norm of $P$, denoted $\mathcal{L}_\infty$. Taking powers of this expression serves as a surrogate loss that captures the system's asymptotic behavior as $T \to \infty$. To test whether this quantity alone governs learning, we applied gradient descent to $\mathcal{L}_\infty$ (with respect to the order parameters). We found, however, that the resulting dynamics were inconsistent with simulation: the dominant eigenvalue descends vertically along the imaginary axis (Fig. 4d; $\alpha = 1$), deviating from the empirical inward trajectory (Fig. 4c).

This discrepancy can be attributed to the fact that although the empirical simulation ultimately minimizes $\mathcal{L}_\infty$, the RNN must also establish short-term control. To capture this transient behavior, we introduce a second term in the surrogate loss. A natural choice is the loss at an early timestep; for $t = 2$, the second term can be written compactly as (see Appendix C.6 for other $t$ values):

$$\mathcal{L}_2 = 2\sigma_{zm}^2 + 2\sigma_{zm} + 6 \tag{13}$$

We now define the full surrogate loss that interpolates between short- and long-horizon terms:

$$\mathcal{L}_{\text{surrogate}} = \alpha \cdot \mathcal{L}_\infty + (1 - \alpha) \cdot \mathcal{L}_2, \quad \alpha \in [0, 1] \tag{14}$$

This blended objective better reflects the dual goals of the second stage: stabilizing the long-term dynamics and achieving short-term immediate control. As expected, setting $\alpha = 0$, optimizing purely for short-term control, drives the dominant eigenvalue upward along the imaginary axis, in direct opposition to the desired asymptotic behavior (Fig. 4d; $\alpha = 0$). In contrast, interpolating with $0 < \alpha < 1$ produces trajectories consistent with empirical observations, where the eigenvalue moves inward toward the unit disk. To further validate this prediction, we initialized multiple RNNs from the same initial condition while varying episode length $T$: shorter episodes pushed the dominant eigenvalue upward, while longer episodes caused it to descend, as predicted by theory (Fig. 4e).

Our theory also explains the two observations described above. First, the zig-zag motion in the empirical trajectory (Fig. 4c) can be attributed to the competing influences of the two loss components, whose gradient directions in the complex plane are nearly opposed. While such zig-zag patterns are well documented in ill-conditioned optimization (e.g., due to large Hessian condition numbers [39, 40]), here they result from a structured conflict between short- and long-term objectives, a hallmark of closed-loop learning. Notably, this conflict is mitigated by adaptive optimizers such as Adam [41] (see Appendix C.7). Second, the divergence between open- and closed-loop test loss may arise because the open-loop RNN, at least initially, does not optimize for the long-term loss component, allowing output weights to grow unboundedly. A full derivation of Stage 2 appears in Appendix C.6.

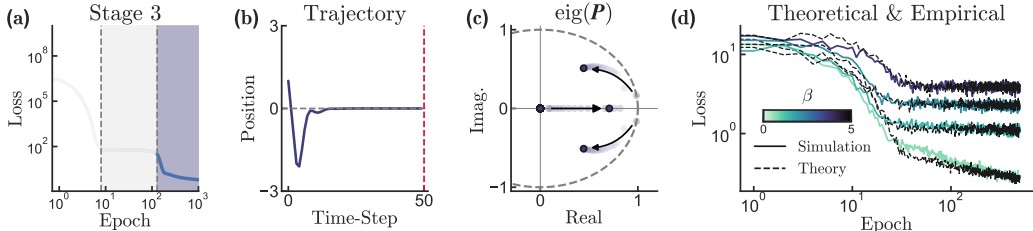

Figure 5: Stage 3 - Policy refinement. **(a)** Final drop in training loss marks the onset of Stage 3. **(b)** Example trajectory of mass position under RNN control at the end of Stage 3, exhibits fast and non-oscillatory dynamics. **(c)** Spectrum of the closed-loop matrix $P$: the dominant complex-conjugate pair contracts further inward, while $\lambda_3$ grows, indicating the emergence of a second slow mode. Arrow indicates trajectory; darker marker denotes final position. **(d)** Training loss during Stage 3 for full high-dimensional model (solid) and the low-dimensional effective model (dashed black), across different regularization strengths $\beta$ (colored), initialized from identical Stage 2 endpoints. The effective model shows excellent agreement with the RNN learning dynamics throughout Stage 3.

**Stage 3 - Policy refinement** Once the closed-loop matrix $P$ stabilizes, the RNN enters a final phase of refinement. Training loss drops again (Fig. 5a), and mass trajectories become fast and non-oscillatory toward the target (Fig. 5b). Notably, the third real eigenvalue $\lambda_3$ increases and begins to influence behavior (Fig. 5c). This marks a shift from Stage 2 where $|\lambda_3| \ll |\lambda_1|$, indicating the emergence of a second slow mode alongside the dominant complex pair. Learning dynamics in this stage are reproduced by directly optimizing the order parameters of the effective model. This is illustrated by varying the control regularization $\beta$ (see Appendix A.2) and comparing the results to the full model. Additional analysis confirms that, despite operating in a stable regime of $P$, Stage 3 learning remains shaped by the same competing objectives as in Stage 2 (Appendix C.8).

## 5 Generalization to multi-frequency tracking tasks

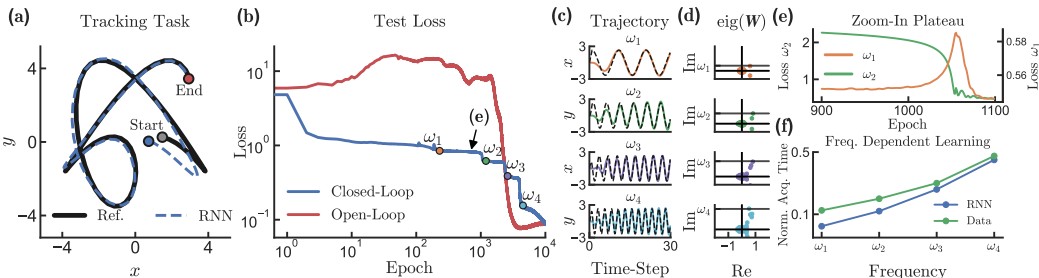

Figure 6: **(a)** Closed-loop RNN trajectory (dashed blue) tracking a 2D multi-frequency target (solid black). **(b)** Test loss across epochs for closed-loop (blue) and open-loop (red) RNNs. Closed-loop RNN exhibits stepwise drops aligned with acquisition of higher-frequency components ($\omega_1 < \omega_2 < \omega_3 < \omega_4$); open-loop shows a loss peak followed by convergence. **(c)** RNN (solid) and reference (dashed) trajectories for individual frequencies, at epochs marked in (b). **(d)** Eigenvalue spectra of $W$ at the same epochs, revealing frequency-specific adaptations. **(e)** Zoom in on the first plateau in (b), with loss decomposed into $\omega_1$ and $\omega_2$ (right and left axes), showing a trade-off between controlling learned and acquiring new frequencies. **(f)** Frequency-dependent learning: both RNN (blue) and humans (green) acquire higher frequencies more slowly. RNN acquisition times were computed from the markers in (b), normalized by total training time; human data from [42] (see Appendix D).

To test the generality of our findings beyond the simple double integrator task, we trained RNNs on a two-dimensional tracking task inspired by human motor control studies [42, 43]. In this task, the RNN must learn to control a cursor to follow a reference trajectory whose motion is defined by a sum of sinusoids with different temporal frequencies (Fig. 6a; see Appendix D for full details).

Consistent with the simpler task, closed-loop RNNs exhibited stage-like learning dynamics: training progressed through discrete plateaus, with each drop in the loss corresponding to the acquisition of a higher-frequency component (Fig. 6b; colored markers). These transitions were validated by testing the RNN on individual frequency components: Fig. 6c shows that each marked epoch aligns with accurate tracking of a newly acquired frequency, while Fig. 6d reveals corresponding adaptations in the spectrum of $W$ (see also Figs. 18 and 19 in Appendix D). In contrast, open-loop RNNs displayed a test loss peak before convergence, consistent with the double integrator.

Furthermore, the loss plateaus in closed-loop RNNs reflect a trade-off between maintaining control over previously acquired frequencies and learning new ones, which require changes in $W$. This is evident in a crossover, where performance on $\omega_1$ temporarily declines as $\omega_2$ is acquired (Fig. 6e). Finally, the frequency-dependent acquisition times of closed-loop RNNs closely tracked trends observed in human participants performing the same task under mirror-reversal perturbation ([42]; Fig. 6f), suggesting a shared inductive bias favoring the early acquisition of low-frequency components.

## 6 Discussion

Learning in biological agents inherently occurs through closed-loop interactions, where future sensory inputs depend on past outputs (i.e., actions). However, most studies train networks, including RNNs, in open-loop supervised settings that omit this feedback structure. To address this gap, we studied the learning dynamics of RNNs under closed-loop conditions in a simplified setting.

Our key finding is that closed-loop learning is fundamentally distinct from its open-loop counterpart. Using our analytical framework, we show that closed-loop RNNs must balance competing short- and long-term objectives. This trade-off temporarily stalls progress, producing a training loss plateau that resolves only when changes in $W$ support an internal representation of hidden variables, such as velocity or frequency. Once formed, these representations allow the output weights $z$ to grow, enabling further improvement. In contrast, open-loop RNNs permit unchecked growth of $z$ early in training, yielding initial gains but ultimately leading to a loss peak when evaluated under closed-loop, out-of-distribution conditions.

In line with prior work on simplicity bias in neural networks, learning often unfolds by first acquiring simple structures before more complex ones [12, 16, 44, 45]. We observed a similar pattern both in policy (e.g., proportional before PID) and in internal representations (e.g., low before high frequencies). Our closed-loop framework offers a novel insight: stage-like learning arises from the need to balance short-term policy gains with longer-term demands requiring internal models of the environment. This trade-off is reflected in loss plateaus, which mark distinct stages of learning. Finally, consistent with recent findings in open-loop RNNs (i.e., uncorrelated inputs) [18], learning proceeds by first aligning weights to task-relevant directions, followed by scaling. Interestingly, in closed-loop RNNs, this scaling is delayed and emerges only after a sufficient internal representation is formed, as indicated by changes in $W$.

Our work complements recent theoretical advancements aimed at broadening learning theory beyond traditional supervised paradigms [34, 46, 47, 48] and architectures beyond feedforward networks [14, 18, 49, 50, 51, 52]. Closely related studies have investigated learning dynamics under RL rules but typically considered weak agent–environment coupling or were restricted to feedforward architectures [46, 47]. Other studies examined implicit biases in networks trained on linear-quadratic regulator (LQR) tasks but largely relied on full-state feedback and used feedforward architectures [34]. Recent empirical research on RNNs trained with closed-loop feedback highlights their potential to replicate aspects of biological motor learning, yet lacks an analytical characterization of the underlying learning dynamics [53]. Our approach addresses these limitations by examining recurrent networks trained on closed-loop tasks with only partial-state feedback, thereby requiring the RNN to develop internal representations of hidden state variables to achieve task goals.

We acknowledge several limitations. Our analysis employed simplified architectures, linear assumptions, and direct gradient computations to enable tractable mathematical derivations. Extending our results to scenarios involving sparse rewards or approximate gradients, typical of broader reinforcement learning settings, remains an important future direction. Furthermore, the reduced system was accurate only within an episode, and future work could derive effective learning dynamics for these order parameters. Finally, although our findings qualitatively extend to nonlinear RNNs, applying this framework to fully nonlinear agents and environments warrants further investigation.

To conclude, our study highlights the importance of incorporating closed-loop dynamics into models of biological learning and demonstrates the usefulness of low-dimensional effective models in understanding RNNs' learning dynamics. We anticipate that this analytical perspective can inform future studies exploring more complex agent-environment interactions.

## Acknowledgments

This work was partially supported by the Israel Science Foundation (1442/21) and the Human Frontier Science Program (RGP0017/2021), both awarded to OB.

## Code Availability

All code was implemented in Python using PyTorch [54] and is available on GitHub: `https://github.com/yoavger/closed_loop_rnn_learning_dynamics`

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

## Appendix overview

The appendix is organized as follows:

- **Section A** – Supplementary experiments: reproducing our main findings using a continuous-time RNN; assessing the impact of regularization (non-zero $\beta$); demonstrating the spontaneous emergence of low-rank structure in unconstrained RNNs; replicating the open-loop setup results from [18]; using an alternative LQR teacher; extension to a more general family of control problems ($k$-integrator).

- **Section B** – Background on key concepts from control theory.

- **Section C** – Full derivation of the closed-loop matrix $P$ and $P_{\text{eff}}$, including showing their equivalent characteristic polynomials. Effective losses for Stages 1 and 2, along with supplemental analyses.

- **Section D** – Additional implementation and training details for the tracking task, along with supplementary plots.

# A  Supplementary experiments

## A.1   Continuous-time RNN

To align with canonical models in neuroscience, we replicate our main findings using a continuous-time RNN (CTRNN), governed by:

$$\tau\,\dot{\boldsymbol{h}} = -\boldsymbol{h} + \phi\left(\boldsymbol{W}\boldsymbol{h} + \boldsymbol{m}y\right), \quad u = \boldsymbol{z}^\top\boldsymbol{h}$$

The double integrator task is likewise formulated in continuous time and discretized using Euler integration:

$$\boldsymbol{A} = \begin{bmatrix} 1 & \Delta t \\ 0 & 1 \end{bmatrix}, \quad \boldsymbol{B} = \begin{bmatrix} 0 \\ \Delta t \end{bmatrix}$$

We use a fixed step size $\Delta t = 0.1$ and a time constant $\tau = 1$. The resulting closed-loop system matrix takes the form:

$$\boldsymbol{P} = \begin{bmatrix} \boldsymbol{A} & \boldsymbol{B}\boldsymbol{z}^\top \\ \Delta t\,\boldsymbol{m}\boldsymbol{C}\boldsymbol{A} & (1 - \Delta t)\boldsymbol{I} + \Delta t\,\boldsymbol{W} \end{bmatrix}$$

As shown below, the continuous-time RNN exhibits the same stage-wise learning dynamics (Fig. 7a), feedback gain trajectories (Fig. 7b-c), and spectral transitions (Fig. 7d-f) as in the discrete-time case. Stage transitions are marked on the test loss (a) and aligned with the corresponding marker in effective gain (b–c). Note that in continuous-time systems, stability requires all eigenvalues to have negative real parts.

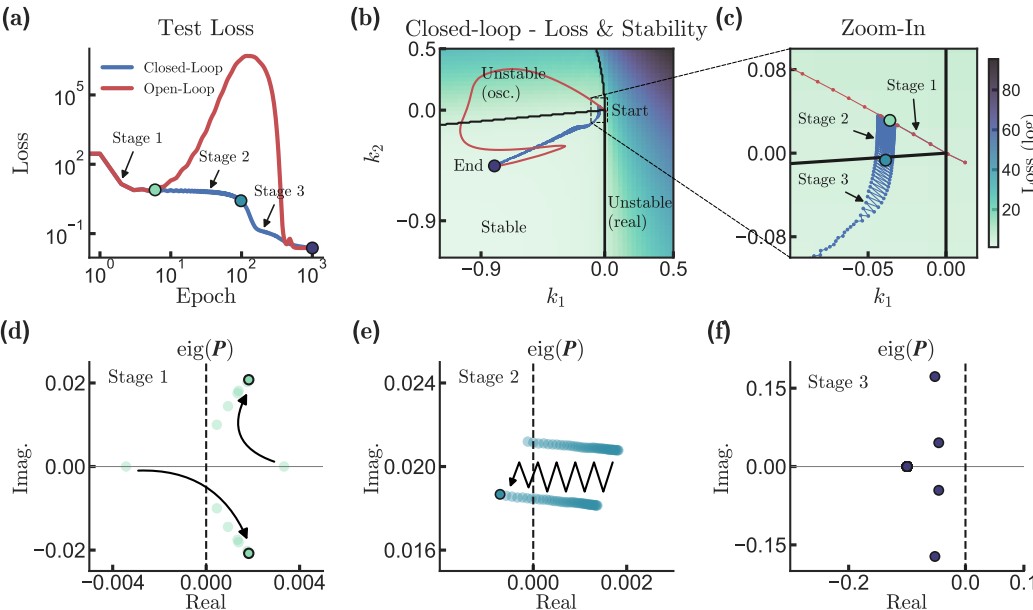

Figure 7:  Continuous-time RNNs show qualitatively similar dynamics to the discrete-time case. **(a)** Test loss reveals three distinct learning stages in the closed-loop RNN; the open-loop RNN shows a sharp loss peak. **(b–c)** Effective feedback gains trace distinct trajectories for each RNN in the $(k_1, k_2)$ plane. Markers denote stage transitions and align with panel (a). **(d–f)** Spectral evolution of the closed-loop matrix $\boldsymbol{P}$ follows the same progression: emergence of complex eigenvalues, zig-zag motion toward stability, and eventual stability. Note that in continuous time, stability requires all eigenvalues to have strictly negative real parts.

## A.2 Regularization

In classical control, penalizing control effort helps balance precision and energy use. Following [34], we set $\beta = 0$ in the main text to isolate learning dynamics without regularization. To assess the impact of regularization, we trained networks with increasing control penalties ($\beta = 0.1, 1$), keeping all other parameters the same.

As shown below, higher $\beta$ increases final train loss (Fig. 8a) and produces more underdamped trajectories (Fig. 8b), reflecting reduced control magnitude. Nonetheless, key features of the learning process, including plateaus (Fig. 8a), convergence to stable feedback gains (Fig. 8c), and the emergence of low-rank structure in the recurrent weights (Fig. 8d-f), remain intact. Thus, we conclude that regularization shifts the final solution but does not disrupt the dynamics of closed-loop learning.

## A.3 Low-rank structure

To characterize the structure of the trained recurrent weights $\boldsymbol{W}$ (initialized with $g = 0.1$), we plot their eigenspectra at convergence (Fig. 8d-f). Across all $\beta$, the spectra are dominated by a single outlier with a compressed bulk near zero, consistent with an emergent low-rank structure [17, 38]. This observation motivates the low rank assumptions used in our analytic derivations.

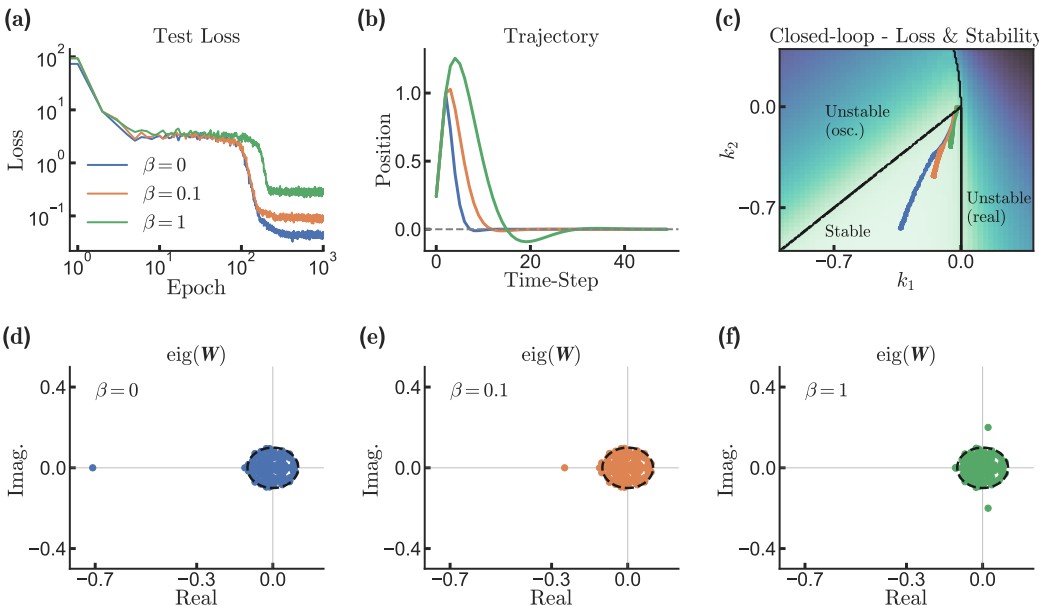

Figure 8: **(a)** Higher $\beta$ increases final test loss while preserving the three-stage learning dynamics. **(b)** Trajectories become more underdamped with increasing $\beta$. **(c)** Effective feedback gains $(k_1, k_2)$ converge to the stable region. **(d-f)** Eigenspectra of $\boldsymbol{W}$ show consistent low rank structure across $\beta$.

## A.4   Open-loop dynamics

In the special case where the RNN has linear activation and is trained in an open-loop setup, the learning task reduces to imitating a linear temporal filter defined by a teacher network. As recently analyzed by [18], this setup admits a detailed analytical treatment. In particular, when initialized with small weights, gradient descent drives the emergence of a pair of outlier eigenvalues in the recurrent matrix $W$ that escape the spectral bulk and converge toward a complex-conjugate pair

$$\lambda_\star = 1 - c_\star \pm i\omega_\star,$$

where $c_\star$ and $\omega_\star$ are determined by the teacher RNN. Although our main text focuses on the less understood closed-loop learning regime, we replicate here the core findings of [18] for completeness (Fig. 9). Following [18], we use a continuous-time RNN (see A.1). In this case, the filter induced by the teacher can be written explicitly as

$$f_\star(t) = z^\top e^{(-I+W)t} m,$$

where $W$, $m$, and $z$ denote the teacher's recurrent matrix, input vector, and output vector, respectively. We then train student RNNs with increasing initial connectivity strength $g$ to imitate this response, using Gaussian white noise as input, exactly matching the open-loop setup described in the main text.

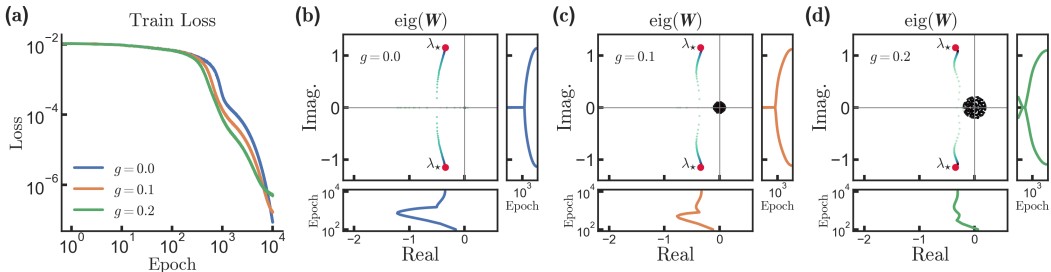

Figure 9: **(a)** RNN student training loss for initial connectivity strengths $g = 0.0, 0.1, 0.2$. Higher $g$ leads to faster convergence. **(b–d)** Training induces a pair of complex-conjugate outlier eigenvalues in the student RNN's recurrent matrix $W$, which escape the bulk and converge to the target $\lambda_\star$ (red dots) specified by the teacher, consistent with [18]. Gradient coloring reflects training epoch, with darker colors indicating later times. Side and bottom panels show the real and imaginary projections of the leading eigenvalues over training epochs. Note that the majority of the bulk are effectively unchanged.

### A.5 LQR teacher

The double-integrator task analyzed in the main text admits an analytically optimal Linear–Quadratic Regulator (LQR) policy (see Appendix B). To test whether the open-loop learning dynamics depend on the teacher, we replaced the RNN teacher used in the main text with this optimal LQR controller and trained the student RNN while the LQR governed the environment. As shown in Fig. 10, the resulting loss trajectory closely matches that obtained with the RNN teacher, with the open-loop dynamics still showing a sharp loss peak midway through training.

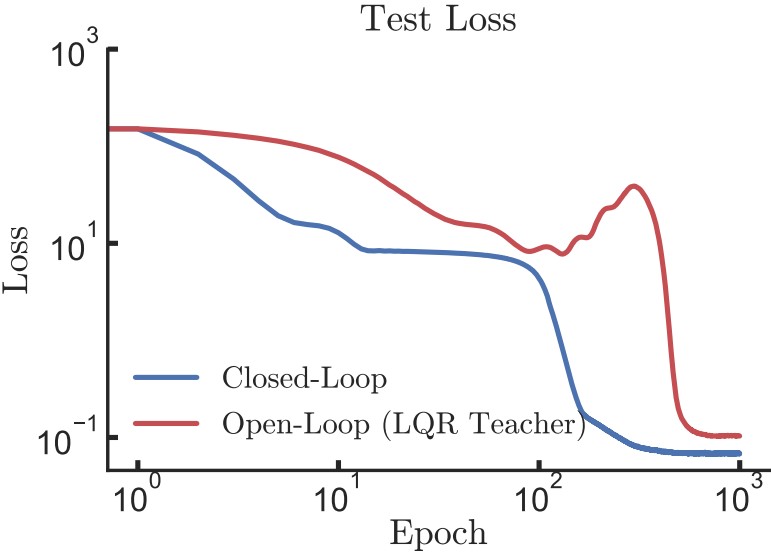

Figure 10: Open-loop learning with an LQR teacher reproduces the transient loss peak.

### A.6 K-integrator task

To test the generality of our findings, we extended the double-integrator setup by introducing the discrete-time $k$-integrator family, a cascade of integrators in which $\boldsymbol{A}, \boldsymbol{B}, \boldsymbol{C}$ take more general forms:

$$\boldsymbol{x}_{t+1} = A_k \boldsymbol{x}_t + B \boldsymbol{u}_t, \qquad \boldsymbol{y}_t = C \boldsymbol{x}_t$$

Here $A_k \in \mathbb{R}^{k \times k}$ is upper-triangular with ones on both the main diagonal and the first superdiagonal:

$$(A_k)_{ij} = \begin{cases} 1 & i = j \text{ or } i = j - 1, \\ 0 & \text{otherwise,} \end{cases}$$

the input matrix $B \in \mathbb{R}^{k \times b}$ assigns each of the last $\min(k, b)$ state components a distinct control input:

$$B_{ij} = \begin{cases} 1 & i = k - j + 1, \ \ j \in \{1, \dots, \min(k, b)\}, \\ 0 & \text{otherwise,} \end{cases}$$

and the observation matrix $C \in \mathbb{R}^{c \times k}$ selects the first $c$ state components:

$$C_{ij} = \begin{cases} 1 & i = j, \ \ i \in \{1, \dots, c\}, \\ 0 & \text{otherwise.} \end{cases}$$

This formulation generalizes the double-integrator task analyzed in the main text ($k = 2$, $b = c = 1$) to arbitrary order and dimensionality, allowing systematic variation of controllability ($b$) and observability ($c$). As shown in Fig. 11, the qualitative gap between closed- and open-loop learning persists across this broader class of systems. In partially observable settings ($c < k$), open-loop RNNs exhibit transient loss peaks and sometimes fail to converge, whereas closed-loop RNNs show a plateau-like trajectory before reaching convergence. Furthermore, under full-state feedback ($c = k$), both training modes behave similarly.

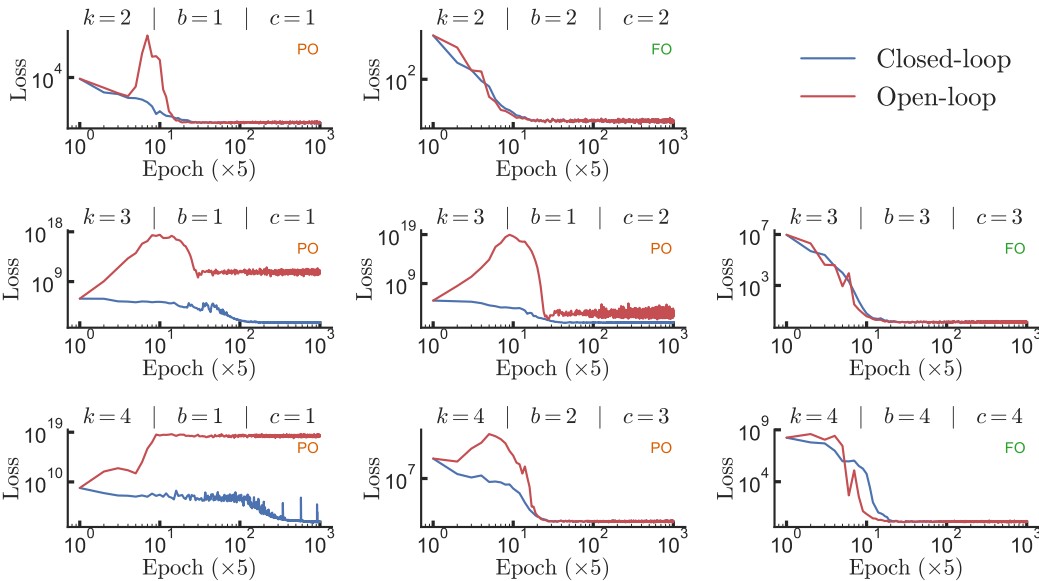

Figure 11: **Open- vs closed-loop learning across the $k$-integrator family.** Training loss for closed- (blue) and open-loop (red) RNNs across varying system order ($k$), number of actuators ($b$), and observed variables ($c$). Partial observability (PO, orange) yields sharp loss peaks and occasional non-convergence in open-loop RNNs, while closed-loop RNNs exhibit plateau dynamics before converging. Under full observability (FO, green), both regimes show nearly identical dynamics.

# B Control theory

In classical control theory, it is standard to treat control, estimation, feedback, and system identification as separate and sequential design problems. In contrast, in our case, the closed-loop RNN must implicitly and simultaneously solve all of these problems during learning. For completeness, we provide the reader with a brief overview of each component.

## B.1 Linear-Quadratic Regulator (LQR)

A fundamental instance of optimal control is the Linear-Quadratic Regulator (LQR) [55], where the system dynamics are linear and the cost function is quadratic. The problem is formulated as (for finite horizon)

$$\min_{\boldsymbol{u}_{0:T}} \sum_{t=0}^{T} \left( \boldsymbol{x}_t^\top \boldsymbol{Q}\, \boldsymbol{x}_t + \boldsymbol{u}_t^\top \boldsymbol{R}\, \boldsymbol{u}_t \right), \quad \text{s.t.} \quad \boldsymbol{x}_{t+1} = \boldsymbol{A}\, \boldsymbol{x}_t + \boldsymbol{B}\, \boldsymbol{u}_t + \boldsymbol{w}_t$$

where $\boldsymbol{x}_t \in \mathbb{R}^n$ is the state, $\boldsymbol{u}_t \in \mathbb{R}^m$ is the control input, and $\boldsymbol{w}_t$ is process noise. The matrices $\boldsymbol{A}$ and $\boldsymbol{B}$ describe the system dynamics, while $\boldsymbol{Q}$ and $\boldsymbol{R}$ are positive semi-definite cost matrices. It is well known that if $\boldsymbol{A}$, $\boldsymbol{B}$, $\boldsymbol{Q}$, and $\boldsymbol{R}$ are known in advance, then the optimal control law takes the form of a linear state feedback:

$$\boldsymbol{u}_t = -\boldsymbol{K}\, \boldsymbol{x}_t$$

where the gain matrix $\boldsymbol{K}$ is given by

$$\boldsymbol{K} = \left( \boldsymbol{R} + \boldsymbol{B}^\top \boldsymbol{M} \boldsymbol{B} \right)^{-1} \boldsymbol{B}^\top \boldsymbol{M} \boldsymbol{A}$$

and $\boldsymbol{M}$ is given by analytically solving the discrete algebraic Riccati equation,

$$\boldsymbol{M} = \boldsymbol{Q} + \boldsymbol{A}^\top \boldsymbol{M} \boldsymbol{A} - \boldsymbol{A}^\top \boldsymbol{M} \boldsymbol{B} \left( \boldsymbol{R} + \boldsymbol{B}^\top \boldsymbol{M} \boldsymbol{B} \right)^{-1} \boldsymbol{B}^\top \boldsymbol{M} \boldsymbol{A}$$

## B.2 Kalman estimation

A common extension of LQR, particularly relevant to biological learning, arises when the state $\boldsymbol{x}_t$ is not directly observable. Instead, we obtain noisy measurements,

$$\boldsymbol{y}_t = \boldsymbol{C}\, \boldsymbol{x}_t + \boldsymbol{v}_t$$

where $\boldsymbol{v}_t$ is measurement noise. By the separation principle, estimation and control can be designed independently, first estimating $\boldsymbol{x}_t$ and then applying the same state-feedback law described above. A standard choice for estimation is the Kalman filter [56], which updates the state estimate $\hat{\boldsymbol{x}}_t$ as

$$\hat{\boldsymbol{x}}_{t+1} = \boldsymbol{A}\, \hat{\boldsymbol{x}}_t + \boldsymbol{B}\, \boldsymbol{u}_t + \boldsymbol{L}_t \left( \boldsymbol{y}_t - \boldsymbol{C}\, \hat{\boldsymbol{x}}_t \right)$$

The Kalman gain $\boldsymbol{L}_t$ is chosen to minimize estimation error, with the gain and covariance update equations:

$$\boldsymbol{L}_t = \boldsymbol{A}\, \Sigma_t \boldsymbol{C}^\top \left( \boldsymbol{C}\, \Sigma_t\, \boldsymbol{C}^\top + \boldsymbol{V} \right)^{-1}, \quad \Sigma_{t+1} = \left( \boldsymbol{A} - \boldsymbol{L}_t \boldsymbol{C} \right) \Sigma_t \boldsymbol{A}^\top + \boldsymbol{W}$$

Here, $\Sigma_t$ quantifies uncertainty in the state estimate, $\boldsymbol{V}$ is the measurement noise covariance, and $\boldsymbol{W}$ is the process noise covariance. The estimated $\hat{\boldsymbol{x}}_t$ replaces $\boldsymbol{x}_t$ in the LQR control law, known as LQG (Linear-Quadratic-Gaussian) control, ensuring optimality under partial observability.

## B.3 System identification

The results above assume that the system matrices are known in advance, enabling optimal estimation and control. However, in many practical settings—including our case—these matrices must be learned from observed data. A common approach is to treat system identification and control design as separate steps. First, the system identification problem is solved to estimate a nominal model, specifically the matrices $\boldsymbol{A}$ and $\boldsymbol{B}$, from data:

$$\min_{\boldsymbol{A}, \boldsymbol{B}} \sum_{t=0}^{N-1} \| \boldsymbol{x}_{t+1} - \boldsymbol{A} \boldsymbol{x}_t - \boldsymbol{B} \boldsymbol{u}_t \|^2$$

Once the nominal model is obtained, it can be used to compute the optimal controller $\boldsymbol{K}$ using the LQR formulation.

### B.4 Closed-loop stability of the double integrator

A fundamental requirement in feedback control is ensuring closed-loop stability, as instability leads to unbounded state growth. The system is stable if the eigenvalues of the closed-loop matrix,

$$M_{\text{cl}} = A - BK$$

lie within the unit disk (for discrete time systems), ensuring that $x_t$ asymptotically converges to zero in the absence of process noise. In our task environment, the system dynamics are given by the discrete-time double integrator:

$$A = \begin{bmatrix} 1 & 1 \\ 0 & 1 \end{bmatrix}, \quad B = \begin{bmatrix} 0 \\ 1 \end{bmatrix}$$

For a feedback policy of the form $u_t = -Kx_t$, where $K \in \mathbb{R}^{1 \times 2}$, the closed-loop system matrix is given by:

$$M_{\text{cl}} = A - BK = \begin{bmatrix} 1 & 1 \\ 0 & 1 \end{bmatrix} - \begin{bmatrix} 0 \\ 1 \end{bmatrix} \begin{bmatrix} k_1 & k_2 \end{bmatrix} = \begin{bmatrix} 1 & 1 \\ -k_1 & 1 - k_2 \end{bmatrix}$$

To determine stability, we compute the eigenvalues of $M_{\text{cl}}$ by solving its characteristic polynomial:

$$\chi_{M_{\text{cl}}}(\lambda) = \det(\lambda I - M_{\text{cl}}) = \lambda^2 - (2 - k_2)\lambda + (1 - k_2 + k_1)$$

which are given by the quadratic formula:

$$\lambda_{\pm} = \frac{(2 - k_2) \pm \sqrt{(k_2 - 2)^2 - 4(1 - k_2 + k_1)}}{2}$$

Stability of the system requires that both eigenvalues lie strictly within the unit circle, i.e., $|\lambda_{\pm}| < 1$. This condition partitions the $(k_1, k_2)$ parameter space into three distinct regimes (see Fig. 12a). The blue region corresponds to stable dynamics, where both eigenvalues are inside the unit circle. The orange region denotes oscillatory instability, in which the eigenvalues are complex conjugates with magnitude exceeding one. The green region corresponds to real instability, where one real eigenvalue satisfies $|\lambda| \geq 1$.

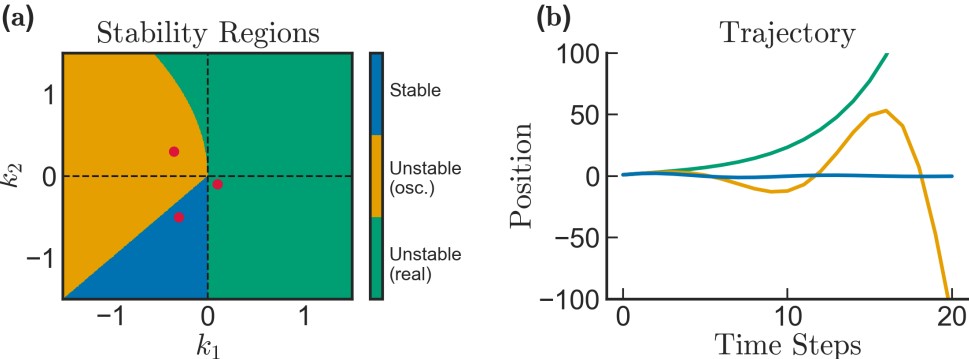

Figure 12: **(a)** Stability regions in the $(k_1, k_2)$ space. The green region represents real unstable modes, the orange region corresponds to unstable oscillatory dynamics, and the blue region indicates stable dynamics. The red dots mark specific parameter choices for which trajectories are simulated. **(b)** Example trajectories of the first component of the state (position) corresponding to the red dots in (a), illustrating how different policies impact the closed-loop system behavior.

## C  Coupled system $P$

### C.1  $P$ matrix derivation

The environment dynamics are given by the linear state-space equations in Eq. 1 and Eq. 2, and the RNN dynamics follow the update rule in Eq. 4. Assuming the RNN takes a linear activation function, we derive below the combined update equations that define the full closed-loop dynamics matrix $P$, as introduced in the main text (Eq. 7).

$$
\begin{aligned}
x_{t+1} &= A x_t + B u_t = A x_t + B\left(z^\top h_t\right) \\
&= A x_t + B z^\top h_t,
\end{aligned}
$$

$$
\begin{aligned}
h_{t+1} &= W h_t + m\, y_{t+1} = W h_t + m\left(C x_{t+1}\right) = W h_t + m\, C\left(A x_t + B z^\top h_t\right) \\
&= m\, C\, A\, x_t + \left(m \overbrace{C B}^{=0} z^\top + W\right) h_t \\
&= m\, C\, A\, x_t + W h_t
\end{aligned}
$$

Or in matrix form:

$$
s_{t+1} = P\, s_t, \quad \text{where} \quad
P = \begin{bmatrix}
\overbrace{A}^{\text{env. dynamics}} & \overbrace{B z^\top}^{\text{control output}} \\
\underbrace{m\, C\, A}_{\text{input feedback}} & \underbrace{W}_{\text{RNN dynamics}}
\end{bmatrix}
$$

### C.2  Characteristic polynomial of $P$ for general $W$

To analyze the stability of the closed-loop system, we examine the eigenvalues of $P$, which satisfy

$$
\det\left(P - \lambda I_{2+N}\right) = 0
$$

Because $P$ has a distinct block structure, we can apply Schur's determinant identity:

$$
\det\begin{pmatrix} A & B \\ C & D \end{pmatrix} = \det(A)\, \det\left(D - C\, A^{-1} B\right)
$$

Applying this to $P - \lambda I_{N+2}$ gives:

$$
\det\left(P - \lambda I_{2+N}\right) = \det\begin{pmatrix} A - \lambda I_2 & B z^\top \\ m\, C\, A & W - \lambda I_N \end{pmatrix} = \det\left(\begin{pmatrix} 1-\lambda & 1 \\ 0 & 1-\lambda \end{pmatrix}\right) \times
$$

$$
\det\left(W - \lambda I_N - \begin{pmatrix} m_1 & m_1 \\ m_2 & m_2 \\ \vdots & \vdots \\ m_N & m_N \end{pmatrix} \times \begin{pmatrix} 1-\lambda & 1 \\ 0 & 1-\lambda \end{pmatrix}^{-1} \times \begin{pmatrix} 0 & 0 & \cdots & 0 \\ z_1 & z_2 & \cdots & z_N \end{pmatrix}\right)
$$

The determinant of the $2 \times 2$ block is

$$
\det\begin{pmatrix} 1-\lambda & 1 \\ 0 & 1-\lambda \end{pmatrix} = (1-\lambda)^2
$$

With further simplification, this factorization leads to

$$
(1-\lambda)^2 \det\left(W + \frac{\lambda}{(1-\lambda)^2}\, m\, z^\top - \lambda I_N\right) = 0
$$

Thus, the eigenvalues of $P$ are given by the solutions to

$$
(1-\lambda)^2 = 0 \quad \text{or} \quad \det\left(W + \tfrac{\lambda}{(1-\lambda)^2}\, m z^\top - \lambda I_N\right) = 0,
$$

yielding a self-consistent eigenvalue equation in $\lambda$, which defines the spectrum of $P$.

## C.3 Rank-one connectivity

Next, we consider the case where $\boldsymbol{W}$ is low-rank, specifically rank-one, and can be expressed as an outer product:

$$\boldsymbol{W} = \boldsymbol{u}\,\boldsymbol{v}^\top$$

where $\boldsymbol{u}, \boldsymbol{v} \in \mathbb{R}^{N \times 1}$. In this case, the second term of the characteristic equation reduces to:

$$\det\left(\boldsymbol{u}\,\boldsymbol{v}^\top + \tfrac{\lambda}{(1-\lambda)^2}\,\boldsymbol{m}\,\boldsymbol{z}^\top - \lambda\,\boldsymbol{I}_N\right) = 0$$

Noting that the matrix $\boldsymbol{u}\,\boldsymbol{v}^\top + \tfrac{\lambda}{(1-\lambda)^2}\,\boldsymbol{m}\,\boldsymbol{z}^\top$ is at most rank-2, we introduce the rank-2 matrices:

$$\boldsymbol{U} = \begin{pmatrix} | & | \\ \boldsymbol{u} & \boldsymbol{m} \\ | & | \end{pmatrix} \quad (N \times 2), \qquad \boldsymbol{V}^\top = \begin{pmatrix} \boldsymbol{v}^\top \\ \dfrac{\lambda}{(1-\lambda)^2}\,\boldsymbol{z}^\top \end{pmatrix} \quad (2 \times N)$$

so that:

$$\boldsymbol{U}\,\boldsymbol{V}^\top = \boldsymbol{u}\,\boldsymbol{v}^\top + \frac{\lambda}{(1-\lambda)^2}\,\boldsymbol{m}\,\boldsymbol{z}^\top$$

Applying the determinant lemma for rank-2 matrices, we have:

$$\det(-\lambda\,\boldsymbol{I}_N + \boldsymbol{U}\,\boldsymbol{V}^\top) = (-\lambda)^n \det\left(\boldsymbol{I}_2 - \frac{1}{\lambda}\boldsymbol{V}^\top\boldsymbol{U}\right)$$

where:

$$\boldsymbol{I}_2 - \frac{1}{\lambda}\boldsymbol{V}^\top\boldsymbol{U} = \begin{pmatrix} 1 - \frac{1}{\lambda}\sigma_{vu} & -\frac{1}{\lambda}\sigma_{vm} \\ -\frac{1}{(1-\lambda)^2}\sigma_{zu} & 1 - \frac{1}{(1-\lambda)^2}\sigma_{zm} \end{pmatrix}$$

Taking the determinant and setting it to zero yields the characteristic polynomial:

$$\chi_{\boldsymbol{P}}(\lambda) = \lambda^3 + (-\sigma_{vu} - 2)\lambda^2 + (2\sigma_{vu} - \sigma_{zm} + 1)\lambda + (\sigma_{vu}\,\sigma_{zm} - \sigma_{vu} - \sigma_{zu}\,\sigma_{vm})$$

a cubic equation as defined in the main text Eq. 9. The resulting cubic equation is fully governed by four scalar quantities, pairwise inner products between RNN vectors, and can be viewed as a low-dimensional representation of the full $\boldsymbol{P}$ system.

## C.4  Low-dimensional effective system

Under the simplifying assumption that the recurrent weights are rank-1, the hidden state $h_t$ evolves within the subspace spanned by $m$ and $u$, and can be written as

$$h_t = \kappa_t^{(m)} \, m + \kappa_t^{(u)} \, u$$

Here, $\kappa_t^{(m)}$ and $\kappa_t^{(u)}$ define effective coordinates for the hidden state, which we collect into a reduced four-dimensional system state:

$$s_t^{(\text{eff})} = \begin{bmatrix} x_t^{(1)} \\ x_t^{(2)} \\ \kappa_t^{(m)} \\ \kappa_t^{(u)} \end{bmatrix} \in \mathbb{R}^4$$

The environment evolves according to

$$x_{t+1}^{(1)} = x_t^{(1)} + x_t^{(2)}, \qquad x_{t+1}^{(2)} = x_t^{(2)} + \sigma_{zm} \, \kappa_t^{(m)} + \sigma_{zu} \, \kappa_t^{(u)}$$

and the RNN hidden state according to

$$h_{t+1} = mCAx_t + uv^\top h_t,$$

with $CA = [1 \ \ 1]$, yielding

$$\kappa_{t+1}^{(m)} = x_t^{(1)} + x_t^{(2)}, \qquad \kappa_{t+1}^{(u)} = \sigma_{vm} \, \kappa_t^{(m)} + \sigma_{vu} \, \kappa_t^{(u)}$$

Together, these updates define a reduced effective system

$$s_{t+1}^{(\text{eff})} = P_{\text{eff}} \, s_t^{(\text{eff})}, \qquad P_{\text{eff}} = \begin{bmatrix} 1 & 1 & 0 & 0 \\ 0 & 1 & \sigma_{zm} & \sigma_{zu} \\ 1 & 1 & 0 & 0 \\ 0 & 0 & \sigma_{vm} & \sigma_{vu} \end{bmatrix}$$

The characteristic polynomial of $P_{\text{eff}}$ is

$$\chi_{P_{\text{eff}}}(\lambda) = \lambda^3 + (-\sigma_{vu} - 2)\lambda^2 + (2\sigma_{vu} - \sigma_{zm} + 1)\lambda + \left(\sigma_{vu}\sigma_{zm} - \sigma_{vu} - \sigma_{zu}\sigma_{vm}\right),$$

which exactly matches the characteristic polynomial of the full matrix $P$ (Eq. 9). This confirms spectral equivalence, in which $P_{\text{eff}}$ preserves the eigenvalues and thus captures the essential dynamics of the full system.

## C.5 Stage 1

In the absence of initial connectivity, we have $\boldsymbol{W} = \boldsymbol{u}\boldsymbol{v}^\top = 0$ (equivalent to setting $g = 0$), the characteristic equation Eq. 9 reduces to:

$$\chi_{\boldsymbol{P}}(\lambda) = \lambda^2 - 2\lambda + (1 - \sigma_{\boldsymbol{zm}})$$

With eigenvalues given by

$$\lambda = 1 \pm \sqrt{\sigma_{\boldsymbol{zm}}}$$

Note that the square root is real if $\sigma_{\boldsymbol{zm}} \geq 0$, and imaginary if $\sigma_{\boldsymbol{zm}} < 0$, in which case the system exhibits complex-conjugate eigenvalues.

**System stability**    We found that the system has two non-trivial eigenvalues, given by $\lambda = 1 \pm \sqrt{\sigma_{\boldsymbol{zm}}}$. Since $|1 + \sqrt{\sigma_{\boldsymbol{zm}}}| \geq 1$, the spectral radius of $\rho(\boldsymbol{P}) > 1$ and therefore the system is unstable. If $\sigma_{\boldsymbol{zm}} < 0$, the eigenvalues are complex with a magnitude exceeding 1, resulting in unbounded oscillations. If $\sigma_{\boldsymbol{zm}} = 0$, the eigenvalues are precisely 1, implying marginal instability. If $\sigma_{\boldsymbol{zm}} > 0$, a single eigenvalue is real and greater than 1, leading to exponential divergence. Overall, the system's behavior either grows exponentially or oscillates with increasing amplitude, depending on the sign of $\sigma_{\boldsymbol{zm}}$.

**Loss approximation**    The loss used to train the RNN agent is given by

$$\mathcal{L} = \frac{1}{T} \sum_{t=1}^{T} \|\boldsymbol{x}_t - \boldsymbol{x}^*\|^2 = \frac{1}{T} \sum_{t=1}^{T} \left[ \left(x_t^{(1)}\right)^2 + \left(x_t^{(2)}\right)^2 \right]$$

We seek a closed-form expression for $\mathcal{L}$ in terms of $\sigma_{\boldsymbol{zm}}$. As $T$ grows large, any mode with an eigenvalue $|\lambda| < 1$ decays to zero, and in the limit $T \to \infty$, the influence of the initial state becomes negligible. We consider two cases:

1. $\sigma_{\boldsymbol{zm}} > 0$: The system has one dominant eigenvalue $\lambda_1 > 1$, with the state evolving as $\alpha_1 \mathbf{v}_1 \lambda_1^t$, where $\boldsymbol{q}_1$ is the corresponding eigenvector and $\alpha_1$ is the projection coefficient of the initial state. The loss simplifies to

$$\mathcal{L} \approx \alpha_1^2 \|\boldsymbol{q}_1\|^2 \sum_{t=0}^{T} \lambda_1^{2t}$$

Recognizing this as a geometric series with ratio $\lambda_1^2$ and ignoring the initial state contribution, we obtain

$$\mathcal{L} \approx \frac{1 - \lambda_1^{2(T+1)}}{1 - \lambda_1^2}$$

2. $\sigma_{\boldsymbol{zm}} < 0$: The system has complex-conjugate eigenvalues with

$$|\lambda_1|^2 = 1 + |\sigma_{\boldsymbol{zm}}|$$

Decomposing the coefficient $\alpha_1$ into real and imaginary parts, $\alpha_1 = a + ib$, the state evolves as

$$\boldsymbol{x}_t = a \cos(t\sqrt{|\sigma_{\boldsymbol{zm}}|}) + b \sin(t\sqrt{|\sigma_{\boldsymbol{zm}}|}),$$

which expands to

$$\boldsymbol{x}_t^2 = a^2 \cos^2(t\sqrt{|\sigma_{\boldsymbol{zm}}|}) + b^2 \sin^2(t\sqrt{|\sigma_{\boldsymbol{zm}}|}) + 2ab \cos(t\sqrt{|\sigma_{\boldsymbol{zm}}|}) \sin(t\sqrt{|\sigma_{\boldsymbol{zm}}|})$$

Using $\cos^2 \theta + \sin^2 \theta = 1$, averaging out oscillatory terms over time, and ignoring initial state contributions, the loss simplifies to

$$\mathcal{L} \approx \frac{1 - |\lambda_1|^{2(T+1)}}{1 - |\lambda_1|^2}$$

## C.6 Stage 2

In this stage, the recurrent weights $\boldsymbol{W} = \boldsymbol{u}\boldsymbol{v}^\top$ evolve, and we must consider the full characteristic cubic equation shown in the main text:

$$\chi_{\boldsymbol{P}}(\lambda) = \lambda^3 + (-\sigma_{\boldsymbol{vu}} - 2)\lambda^2 + (2\sigma_{\boldsymbol{vu}} - \sigma_{\boldsymbol{zm}} + 1)\lambda + (\sigma_{\boldsymbol{vu}}\,\sigma_{\boldsymbol{zm}} - \sigma_{\boldsymbol{vu}} - \sigma_{\boldsymbol{zu}}\,\sigma_{\boldsymbol{vm}})$$

$\mathcal{L}_\infty$ **approximation**  The characteristic equation above takes the general cubic form:

$$a\lambda^3 + b\lambda^2 + c\lambda + d = 0$$

where the coefficients $a, b, c, d$ are defined above. Denoting the roots as $\lambda_1$, $\lambda_2 = \overline{\lambda_1}$ (complex conjugates), and $\lambda_3 \in \mathbb{R}$, this structure allows us to establish a direct relationship between $|\lambda_1|^2$ and $\lambda_3$ using Vieta's formulas, which relate the polynomial's roots to its coefficients. From Vieta's first formula, the sum of the roots is:

$$\lambda_1 + \lambda_2 + \lambda_3 = -\frac{b}{a}$$

Since $\lambda_1 + \lambda_2 = 2\operatorname{Re}(\lambda_1)$, it follows that:

$$\operatorname{Re}(\lambda_1) = \frac{-\frac{b}{a} - \lambda_3}{2}$$

Vieta's second formula gives the sum of the products of root pairs:

$$\lambda_1\lambda_2 + \lambda_1\lambda_3 + \lambda_2\lambda_3 = \frac{c}{a}$$

Using $\lambda_1\lambda_2 = |\lambda_1|^2$ and $\lambda_1 + \lambda_2 = 2\operatorname{Re}(\lambda_1)$, we have:

$$|\lambda_1|^2 + 2\lambda_3 \operatorname{Re}(\lambda_1) = \frac{c}{a}$$

Substituting the expression for $\operatorname{Re}(\lambda_1)$, we obtain:

$$|\lambda_1|^2 + 2\lambda_3\left(\frac{-\frac{b}{a} - \lambda_3}{2}\right) = \frac{c}{a}$$

Simplifying gives:

$$|\lambda_1|^2 = \frac{c}{a} + \frac{b}{a}\lambda_3 + \lambda_3^2$$

Substituting the coefficients in terms of the order parameters reproduces Eq. 11 of the main text. This yields a direct relationship between the magnitude of the complex eigenvalues and the real root $\lambda_3$, expressed entirely through the four scalar overlap parameters.

**Perturbation approximation of $\lambda_3$**  We now derive a closed-form approximation for the real eigenvalue $\lambda_3$. Since we take $\boldsymbol{W} = \boldsymbol{u}\boldsymbol{v}^\top$, we know that the unperturbed matrix has a single non-zero eigenvalue $\sigma_{\boldsymbol{vu}}$. To quantify how the feedback term modifies this eigenvalue, we apply first-order perturbation theory, treating $\boldsymbol{u}\boldsymbol{v}^\top$ as the unperturbed matrix and $\frac{\lambda}{(1-\lambda)^2}\boldsymbol{m}\boldsymbol{z}^\top$ as a perturbation. Expanding the characteristic equation $\chi_{\boldsymbol{P}}(\lambda) = 0$ around $\lambda = \sigma_{\boldsymbol{vu}}$, and using the first-order correction formula:

$$\lambda_3 \approx \lambda_0 - \frac{\chi_{\boldsymbol{P}}(\lambda_0)}{\chi'_{\boldsymbol{P}}(\lambda_0)} \quad \text{with} \quad \lambda_0 = \sigma_{\boldsymbol{vu}}$$

We obtain:

$$\lambda_3 \approx \sigma_{\boldsymbol{vu}} + \frac{\sigma_{\boldsymbol{zu}}\,\sigma_{\boldsymbol{vm}}}{(\sigma_{\boldsymbol{vu}})^2 - 2\sigma_{\boldsymbol{vu}} - \sigma_{\boldsymbol{zm}} + 1} + \mathcal{O}(\epsilon^2)$$

This expression captures the leading-order shift in the eigenvalue due to the low-rank input–output term, with many terms in the numerator and denominator canceling upon substitution.

$\mathcal{L}_t$ **approximation** To approximate the early-time contribution to the loss, we define a second objective based on powers of the system matrix $\boldsymbol{P}$. Specifically, we extract the upper-left $2 \times 2$ block of $\boldsymbol{P}$ (i.e., $\boldsymbol{A}$ block) and define:

$$\mathcal{L}_t = \sum_{i,j=1}^{2} \left( \boldsymbol{P}_{i,j}^t \right)^2$$

This quantity approximates the expected loss at a single early timestep under random initial conditions. To show this, consider the evolution $\boldsymbol{x}_t = \boldsymbol{P}^t \boldsymbol{x}_0$, with target state $\boldsymbol{x}^* = 0$, the instantaneous loss is quadratic in $\boldsymbol{x}_t$. Since $\boldsymbol{x}_0$ is random and consists of independent, zero-mean components with identical variance, the expected early-time loss simplifies as cross-terms cancel out in expectation. Specifically, after applying $\boldsymbol{P}^t$, the first component of the state satisfies:

$$x_t^{(1)} = \boldsymbol{P}_{11}^t \, x_0^{(1)} + \boldsymbol{P}_{12}^t \, x_0^{(2)}$$

and the expected squared norm is:

$$\mathbb{E}\left[ (x_t^{(1)})^2 \right] = (\boldsymbol{P}_{11}^t)^2 \mathbb{E}[(x_0^{(1)})^2] + (\boldsymbol{P}_{12}^t)^2 \mathbb{E}[(x_0^{(2)})^2]$$

where we used $\mathbb{E}[x_0^{(1)} x_0^{(2)}] = 0$ by independence at $t = 0$. The same argument applies to the second coordinate, yielding:

$$\mathcal{L}_t \propto (\boldsymbol{P}_{11}^t)^2 + (\boldsymbol{P}_{12}^t)^2 + (\boldsymbol{P}_{21}^t)^2 + (\boldsymbol{P}_{22}^t)^2$$

where the proportionality constant depends on the variance of the initial state.

For example, for $t = 1$,

$$\boldsymbol{P}_{1:2,1:2}^1 = \begin{pmatrix} 1 & 1 \\ 0 & 1 \end{pmatrix} \quad \Rightarrow \quad \mathcal{L}_1 = 1^2 + 1^2 + 0^2 + 1^2 = 3$$

for $t = 2$ (Eq. 13),

$$\boldsymbol{P}_{1:2,1:2}^2 = \begin{pmatrix} 1 & 2 \\ \sigma_{zm} & \sigma_{zm} + 1 \end{pmatrix} \quad \Rightarrow \quad \mathcal{L}_2 = 2\sigma_{zm}^2 + 2\sigma_{zm} + 6$$

and for $t = 3$,

$$\boldsymbol{P}_{1:2,1:2}^3 = \begin{pmatrix} \sigma_{zm} + 1 & \sigma_{zm} + 3 \\ \sigma_{vm}\sigma_{zu} + 2\sigma_{zm} & \sigma_{vm}\sigma_{zu} + 3\sigma_{zm} + 1 \end{pmatrix}$$

$$\Rightarrow \quad \mathcal{L}_3 = (\sigma_{zm} + 1)^2 + (\sigma_{zm} + 3)^2 + (\sigma_{vm}\sigma_{zu} + 2\sigma_{zm})^2 + (\sigma_{vm}\sigma_{zu} + 3\sigma_{zm} + 1)^2$$

We therefore used $t = 2$ in the main text (Eq. 13), as it is the lowest order that already includes the order parameters in its expansion. Note that $t = 3$ can also reproduce the results of Fig. 4d, but it requires a different tuning of the $\alpha$ parameters due to the distinct scaling of the loss.

## C.7 SGD vs. Adam

The choice of optimizer influences learning dynamics in Stage 2. With SGD, the training loss shows oscillations due to shifts in the output weights between more and less negative values. This reflects a trade-off between early- and late-episode control.

As illustrated on the right (Fig. 13), the differing curvature along the output weight direction $z$ and the recurrent weight direction $W$ causes updates to alternate between prioritizing short-term performance (green region, lower right) and long-term stability (red region, upper left), producing a characteristic zig-zag trajectory. This trade-off is also evident empirically: more negative output weights improve short-term performance but cause divergence later in the episode (green in Fig. 14a), whereas less negative weights reduce late-episode instability at the expense of higher early error (red in Fig. 14a). The resulting zig-zag pattern in the loss (Fig. 14b) is driven by sharp sign changes in the projected gradient $\nabla z^\top m$ (Fig. 14d).

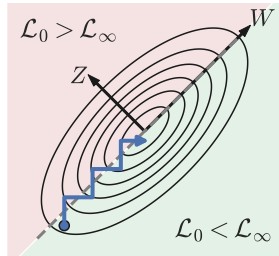

Figure 13: Illustration of zig-zag learning dynamics.

Furthermore, as discussed in the main text, alternating changes in the overlap $z^\top m$ (Fig. 14c) contribute to zig-zag motion of the dominant eigenvalues of $P$ across the complex plane (Fig. 14e).

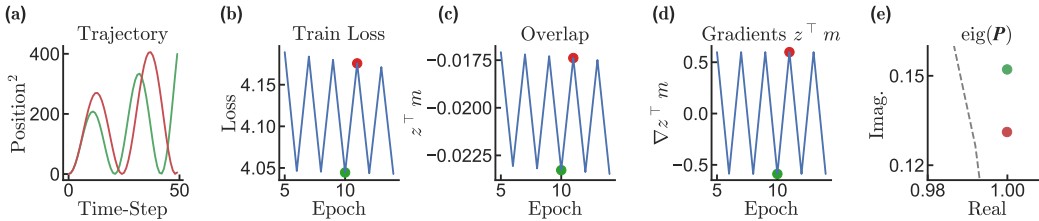

Figure 14: **(a)** Squared position $(x^{(1)})^2$ over time, showing alternating trajectories from epochs with more (red) vs. less (green) negative output weights. **(b)** Zoom-in to the training loss exhibits oscillations during Stage 2. **(c)** Overlap $z^\top m$, alternating between more and less negative values. **(d)** Projected gradient $\nabla z^\top m$, showing sign flips across epochs. **(e)** Dominant eigenvalues of $P$, shifting across the complex plane in response.

In contrast, Adam uses running averages of gradients to stabilize updates, eliminating oscillations and producing smooth convergence during Stage 2 (Fig. 15). Note, however, that while Adam mitigates the loss plateaus and zig-zag patterns observed with SGD, it does not eliminate them entirely. Plateaus may still emerge, typically shorter or requiring mild parameter tuning. In more complex tasks, we find that such plateaus arise more readily, even when using Adam (Appendix D).

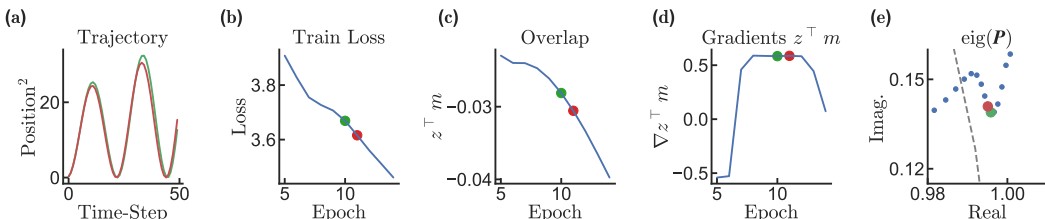

Figure 15: Adam smooths updates and eliminates oscillations, yielding stable convergence during Stage 2.

## C.8 Stage 3

**Loss dynamics remain shaped by competing forces.** Although Stage 3 operates in a stable regime, the final drop in training loss reflects ongoing tradeoffs between early- and late-episode performance, as in Stage 2. This is supported in two ways. First, RNNs initialized within this regime and trained using the open-loop setup exhibit divergent $(k_1, k_2)$ trajectories compared to the closed-loop path (Fig. 16a). Second, decomposition of the total loss into early and late episode segments reveals that improvements in one often coincide with degradation in the other (Fig. 16b). This confirms that closed-loop learning remains uniquely structured, even after reaching the stable regime.

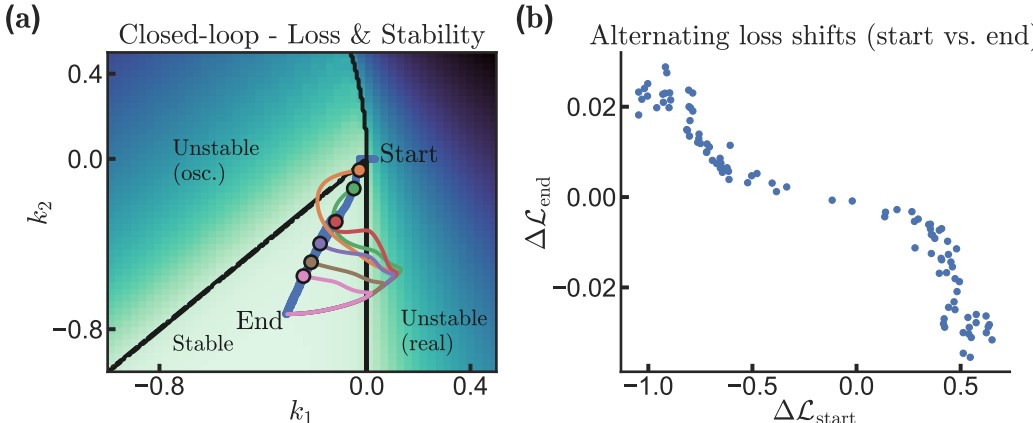

Figure 16: **(a)** Numerical estimation of the effective RNN gain $K_{\text{eff}}$. The blue curve shows the closed-loop evolution during Stage 3. Colored curves depict open-loop training from different initialization points sampled along this trajectory. Despite being initialized in the same stable region, the open-loop paths diverge from the closed-loop trajectory, highlighting their distinct learning dynamics. **(b)** Decomposition of the loss into early and late episode segments. Each point compares the change in early loss $\Delta\mathcal{L}_{\text{start}}$ and late loss $\Delta\mathcal{L}_{\text{end}}$ between consecutive epochs. The negative correlation indicates an alternating pattern: improvements in one segment typically coincide with degradation in the other.

## C.9 Eigenvector–based recovery of $(k_1, k_2)$

Under the rank-1 assumption, the full closed-loop matrix $\boldsymbol{P} \in \mathbb{R}^{(2+N)\times(2+N)}$ reduces exactly to the $4 \times 4$ effective matrix $\boldsymbol{P}_{\text{eff}}$ (see C.4). During Stages 1–2, the spectrum of $\boldsymbol{P}_{\text{eff}}$ is dominated by a complex-conjugate pair $\lambda_1, \bar{\lambda}_1$. We denote the corresponding eigenvector by $\boldsymbol{q}_1 \in \mathbb{C}^4$, and separate it into real and imaginary parts:

$$\boldsymbol{q}_{\text{R}} = \text{Re}(\boldsymbol{q}_1), \qquad \boldsymbol{q}_{\text{I}} = \text{Im}(\boldsymbol{q}_1)$$

These vectors are linearly independent and span the slow invariant subspace. Each eigenvector can be decomposed into an environment projection and a hidden projection by,

$$\boldsymbol{q}_{\text{R}} = [x_{\text{R}}^{(1)},\ x_{\text{R}}^{(2)},\ \kappa_{\text{R}}^{(m)},\ \kappa_{\text{R}}^{(u)}]^\top, \qquad \boldsymbol{q}_{\text{I}} = [x_{\text{I}}^{(1)},\ x_{\text{I}}^{(2)},\ \kappa_{\text{I}}^{(m)},\ \kappa_{\text{I}}^{(u)}]^\top$$

where the first two entries represent the environment projection (position and velocity), and the last two correspond to the projection of the RNN hidden state onto $\boldsymbol{m}$ and $\boldsymbol{u}$. From the second row (i.e., velocity update) of the eigenvalue equation $\boldsymbol{P}_{\text{eff}}\boldsymbol{q}_* = \lambda_1 \boldsymbol{q}_*$, we obtain:

$$(\lambda_1 - 1)\, x_{\text{R}}^{(2)} = k_1 x_{\text{R}}^{(1)} + k_2 x_{\text{R}}^{(2)}, \qquad (\lambda_1 - 1)\, x_{\text{I}}^{(2)} = k_1 x_{\text{I}}^{(1)} + k_2 x_{\text{I}}^{(2)}$$

This defines a linear system for the feedback gains denoted as $\boldsymbol{K}_{\text{exact}}$. Let

$$\boldsymbol{Q} = \begin{bmatrix} x_{\text{R}}^{(1)} & x_{\text{I}}^{(1)} \\ x_{\text{R}}^{(2)} & x_{\text{I}}^{(2)} \end{bmatrix}, \qquad \boldsymbol{u} = (\lambda_1 - 1) \begin{bmatrix} x_{\text{R}}^{(2)} \\ x_{\text{I}}^{(2)} \end{bmatrix},$$

so that

$$\boldsymbol{Q}^\top \boldsymbol{K}_{\text{exact}} = \boldsymbol{u} \quad \Rightarrow \quad \boldsymbol{K}_{\text{exact}} = \boldsymbol{Q}^{-\top} \boldsymbol{u}.$$

Comparison between the analytic $\boldsymbol{K}_{\text{exact}}$ recovery and the numerical estimates of $\boldsymbol{K}_{\text{eff}}$ (main text Eq. 6) shows excellent agreement during Stage 2, with deviations emerging in Stage 3 (see Fig. 17). Note that the analytic recovery relies on the eigenvectors of $\boldsymbol{P}_{\text{eff}}$, which do admit a closed-form expression, though it is lengthy and not written in compact form.

**When is eigenvector-based recovery exact?**   The mapping $\boldsymbol{P} \mapsto (k_1, k_2)$ is *exact* when the closed-loop dynamics are effectively two-dimensional. This condition is met throughout Stages 1–2, and most clearly during Stage 2, where the dynamics enter an oscillatory phase dominated by a conjugate pair. We further know that in this phase, the third eigenvalue can be approximated by $\lambda_3 \approx \sigma_{\boldsymbol{vu}}$ and remains small with $|\lambda_3| \ll |\lambda_1|$.

**Stage 3: correlated third mode**   As training progresses into Stage 3, a third eigenvalue $\lambda_3$ moves toward the unit circle and is no longer negligible, breaking the earlier approximation. However, empirically, we found that this mode remains strongly correlated with the dominant conjugate pair, and the three directions continue to span a nearly rank-2 subspace. In this setting, projecting onto $(k_1, k_2)$ is no longer exact but remains a good approximation due to persistent alignment among the leading modes. Therefore, the effective gains $\boldsymbol{K}_{\text{eff}}$ can still be estimated numerically (see main text).

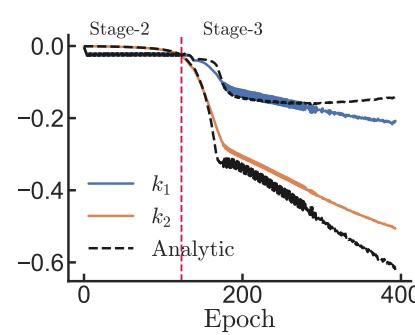

Figure 17: Numerical estimates of the effective feedback gains $\boldsymbol{K}_{\text{eff}}$ during training (solid lines) compared with eigenvector-based recovery $\boldsymbol{K}_{\text{exact}}$ (dashed black). Agreement is exact during Stage 2, and begins to diverge in Stage 3 with the emergence of a third slow mode. The red vertical dashed line denotes the transition from Stage 2 to 3.

**Higher-order regimes**   If three or more slow modes emerge without strong correlation—e.g., a genuine rank-3 structure—no fixed pair $(k_1, k_2)$ can capture the closed-loop behavior. The controller then implements a higher-order policy, and projecting onto $u_t = -k_1 x_t^{(1)} - k_2 x_t^{(2)}$ discards essential structure.

To conclude, exact recovery holds when the dominant conjugate pair is isolated by a spectral gap (i.e., $|\lambda_3| \ll |\lambda_1|$). In Stage 3, approximate recovery remains valid due to residual alignment. Beyond this regime, capturing the full closed-loop dynamics requires additional controller states or alternative representations.

## D  Tracking task

We train RNNs to track two-dimensional target trajectories composed of the sum of two sinusoids along each axis. Below, we describe the dynamics of the target position (reference signal $\boldsymbol{r}$), cursor position $\boldsymbol{x}$, and RNN controller $\boldsymbol{h}$. Following [42, 43], the reference signals were generated as:

$$r_x(t) = a_1 \cos(\omega_1 t + \phi_1) + a_2 \cos(\omega_3 t + \phi_3), \quad r_y(t) = a_1 \cos(\omega_2 t + \phi_2) + a_2 \cos(\omega_4 t + \phi_4)$$

where amplitudes $a_1 = a_2 = 2.31$, the phases $\phi_i$ were sampled uniformly from $[-\pi, \pi]$ at each episode, and the angular frequencies were

$$\omega_1 = 2\pi \times 0.10, \quad \omega_2 = 2\pi \times 0.20, \quad \omega_3 = 2\pi \times 0.30, \quad \omega_4 = 2\pi \times 0.40$$

corresponding to increasing frequencies along each axis. The trajectories had a total duration of 30 seconds, sampled at 10 Hz ($n_{\text{samples}} = 300$), and a linear ramp was applied to the amplitude over the first second.

Each trajectory obeyed a linear state-space model

$$\dot{\boldsymbol{r}}(t) = \boldsymbol{R}\boldsymbol{r}(t), \quad \boldsymbol{R} = \begin{bmatrix} 0 & 1 & 0 & 0 & 0 & 0 & 0 & 0 \\ -\omega_1^2 & 0 & 0 & 0 & 0 & 0 & 0 & 0 \\ 0 & 0 & 0 & 1 & 0 & 0 & 0 & 0 \\ 0 & 0 & -\omega_3^2 & 0 & 0 & 0 & 0 & 0 \\ 0 & 0 & 0 & 0 & 0 & 1 & 0 & 0 \\ 0 & 0 & 0 & 0 & -\omega_2^2 & 0 & 0 & 0 \\ 0 & 0 & 0 & 0 & 0 & 0 & 0 & 1 \\ 0 & 0 & 0 & 0 & 0 & 0 & -\omega_4^2 & 0 \end{bmatrix}$$

with output

$$\boldsymbol{y}_r(t) = \boldsymbol{C}_R \boldsymbol{r}(t), \quad \boldsymbol{C}_R = \begin{bmatrix} 1 & 0 & 1 & 0 & 0 & 0 & 0 & 0 \\ 0 & 0 & 0 & 0 & 1 & 0 & 1 & 0 \end{bmatrix}$$

discretized with time step $\Delta t = 0.1$ seconds as $\boldsymbol{R}_d = \exp(\boldsymbol{R}\Delta t)$.

**Agent and environment**  The agent was modeled by a continuous-time RNN (see A.1) with $N = 100$ hidden units and linear activation, discretized using Euler's method with step size $\Delta t = 0.1$. The RNN had input weight matrix $\boldsymbol{M} \in \mathbb{R}^{N \times 4}$, recurrent weight matrix $\boldsymbol{W} \in \mathbb{R}^{N \times N}$, and output weight matrix $\boldsymbol{Z} \in \mathbb{R}^{N \times 2}$. The environment was a two-dimensional plant. Each axis $(x, \dot{x})$ and $(y, \dot{y})$ followed independent continuous-time dynamics and was simulated using Euler's method:

$$\boldsymbol{x}_{t+1} = \boldsymbol{A}\boldsymbol{x}_t + \boldsymbol{B}u_t, \quad \boldsymbol{A} = \begin{bmatrix} 1 & \Delta t \\ 0 & 1 \end{bmatrix}, \quad \boldsymbol{B} = \begin{bmatrix} 0 \\ \Delta t \end{bmatrix},$$

where $\boldsymbol{x}_t = (x, \dot{x}, y, \dot{y})^\top$ is the full plant state and $\boldsymbol{u}_t \in \mathbb{R}^2$ are the accelerations commanded by the agent. Here, $x$ and $y$ denote the horizontal and vertical positions of a cursor, while $\dot{x}$ and $\dot{y}$ represent its velocities and the agent is tasked with matching the cursor to the time-varying target $(r_x(t), r_y(t))$. The observation matrix $\boldsymbol{C} \in \mathbb{R}^{2 \times 4}$ projects the full plant state onto observed positions:

$$\boldsymbol{C} = \begin{bmatrix} 1 & 0 & 0 & 0 \\ 0 & 0 & 1 & 0 \end{bmatrix}$$

At each time step, the RNN received a 4-dimensional input vector consisting of the observed cursor position and the current target position $\big(x(t), y(t), r_x(t), r_y(t)\big)^\top$, and output a 2-dimensional control vector $\boldsymbol{u}_t = (u_x, u_y)$ corresponding to the commanded accelerations along each axis.

**Closed-loop dynamics**  The full closed-loop system, combining the reference generator, the plant, and the RNN controller, was captured by the block transition matrix:

$$\boldsymbol{P} = \begin{bmatrix} \boldsymbol{R}_d & \boldsymbol{0} & \boldsymbol{0} \\ \boldsymbol{0} & \boldsymbol{A} \oplus \boldsymbol{A} & (\boldsymbol{B} \oplus \boldsymbol{B})\boldsymbol{Z}^\top \\ \Delta t\, \boldsymbol{M}^\top \boldsymbol{C}_R & \Delta t\, \boldsymbol{M}^\top \boldsymbol{C}(\boldsymbol{A} \oplus \boldsymbol{A}) & (1 - \Delta t)\, \boldsymbol{I} + \Delta t\, \boldsymbol{W}^\top \end{bmatrix}$$

where $\oplus$ denotes the block-diagonal sum over the two axes and $\boldsymbol{C}$ maps plant states to observed positions.

**Training**    We trained the RNN controller under two regimes: closed-loop and open-loop. In both cases, training used the Adam optimizer [41] (learning rate $10^{-3}$, batch size 100), minimizing a time-averaged loss over $T = 30$ seconds.

**Closed-loop**    In this regime, the RNN interacted with the environment and was trained to minimize the squared position tracking error:

$$\mathcal{L}_{\text{closed}} = \frac{1}{T} \sum_{t=1}^{T} \|x(t) - r_x(t)\|^2 + \|y(t) - r_y(t))\|^2$$

Target trajectories were generated as described above, with random phases $\phi_i \sim \mathcal{U}(-\pi, \pi)$ resampled each episode. At each timestep, the RNN received the observed cursor position and the current target $(x, y, r_x(t), r_y(t))$, produced a control vector, and influenced the system. The hidden state was reset between episodes. Plant states were clamped to $[-10, 10]$ to prevent divergence. Fixing the input weights $\boldsymbol{M}$ did not affect convergence in this setup.

**Open-loop**    Here, a randomly initialized student RNN was trained to match the outputs of a pre-trained teacher (from the final closed-loop epoch). Both networks received the same input $(x(t), y(t), r_x(t), r_y(t))$, where $(x(t), y(t))$ evolved under teacher control. The student was optimized to minimize the squared difference in outputs:

$$\mathcal{L}_{\text{open}} = \frac{1}{T} \sum_{t=1}^{T} \|\hat{\boldsymbol{u}}_{\text{student}}(t) - \boldsymbol{u}_{\text{teacher}}(t)\|^2$$

This setup only converged when the teacher drove the plant. Using Gaussian white noise in place of state inputs led to unstable or ineffective student policies. In contrast to the closed-loop case, learning the input weights $\boldsymbol{M}$ was essential for the student to match the teacher.

**Loss crossover**    To highlight the trade-off between maintaining control over a previously learned frequency and acquiring a new one, which requires changes in $\boldsymbol{W}$, we tested the RNN from a checkpoint taken during the first loss plateau (marked by **(e)** in Fig. 6b) on individual frequencies. As shown in Fig. 6e, this reveals a crossover where performance on $\omega_1$ temporarily declines as $\omega_2$ is acquired.

**Human data**    To compute the green Data curve in Fig. 6f, we extracted summary data from [42], which reported distinct participants' learning rates across frequencies (lower frequencies acquired more rapidly than higher ones). For the first four frequencies ($\omega_1$–$\omega_4$), we identified the first time block at which the mean orthogonal gain (averaged across participants) exceeded 0.3, and normalized this value by the total duration (20 blocks). Orthogonal gain quantifies the component of motor output aligned with the target direction after mirror reversal.

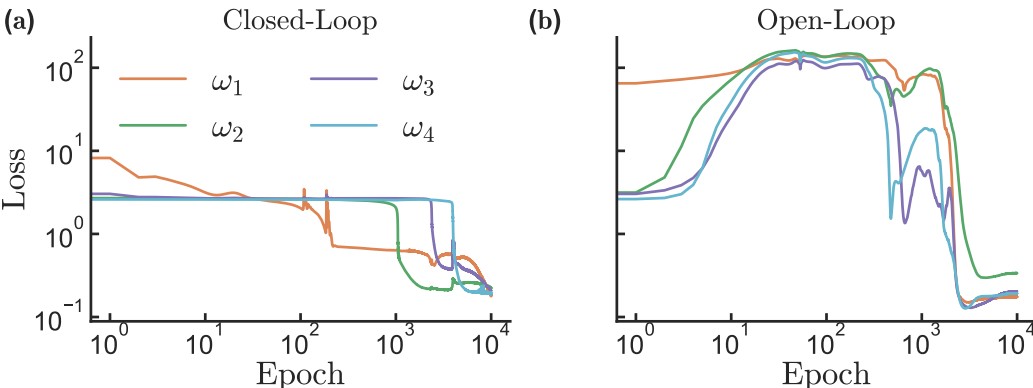

Figure 18: Frequency-specific loss decomposition during training on the multi-frequency tracking task. **(a)** Closed-loop training exhibits stage-like convergence: loss decreases sequentially for each frequency component ($\omega_1$–$\omega_4$), reflecting their progressive acquisition. **(b)** In contrast, open-loop training, loss initially rises, then decreases across all frequencies more uniformly, reflecting less structured acquisition.

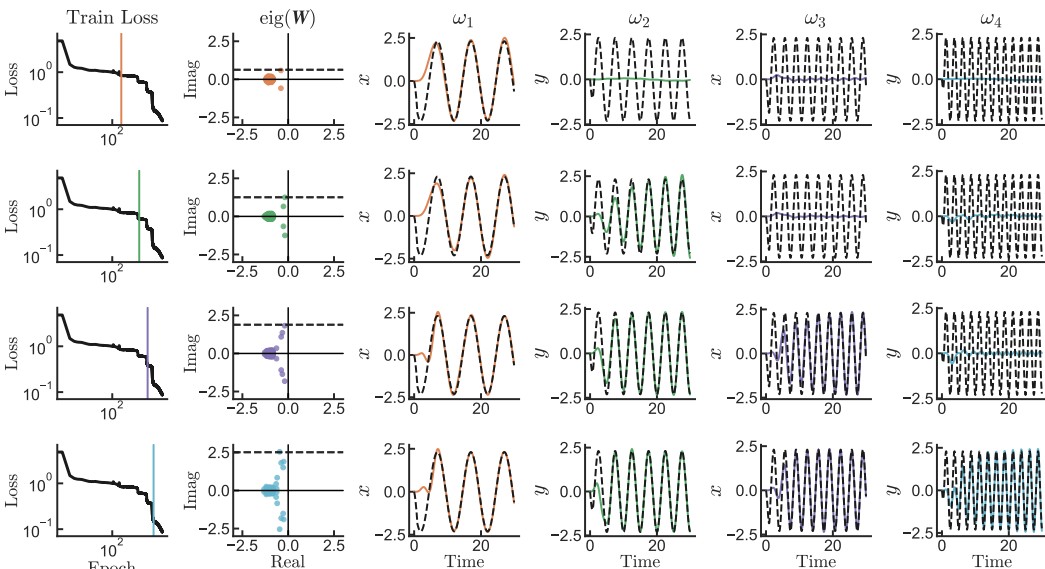

Figure 19: Supplement to Fig. 6, illustrating frequency-specific RNN learning in the tracking task. Each row shows (left to right): closed-loop RNN training loss, eigenvalue spectrum of the recurrent weights $\boldsymbol{W}$, and RNN outputs for isolated frequency components $\omega_1$–$\omega_4$, evaluated at key epochs during closed-loop training (indicated by vertical colored lines the first column). Colored vertical lines match the markers in Fig. 6b.

