# OpenReview forum: "Learning Dynamics of RNNs in Closed-Loop Environments"
_NeurIPS.cc/2025/Conference — NeurIPS 2025 poster_

### Official Review · Reviewer_xBz7 · 2025-06-30

**Clarity:** 2
**Significance:** 3
**Originality:** 3
**Rating:** 5
**Confidence:** 3

**Summary:**

The paper studies the learning dynamics of recurrent neural networks in closed-loop scenarios and compares it to the corresponding ones in open-loop scenarios. It finds that they significantly differ, with open-loop dynamics failing to properly solve the closed-loop problem mid-training. The paper uses a simplified model and task to thoroughly investigate these dynamics theoretically, as well as provides empirical evidence that the findings holds on a (slightly) more complicated task.

**Questions:**

From my understanding, the authors use a network trained on the task as the teacher for the open-loop setting. Why not training it on the **optimal control policy** (which should be tractable). Given that trained networks do not reach 0 loss, their predictions might contain some strange artifacts that affect learning dynamics. Can the authors comment on that and perform, if possible, a quick experiment to verify that?

Can the authors provide intuition on l. 250 and on why low frequencies are learned faster?

Could the analysis provided by the authors be extended to **higher-dimensional systems**? Additionally, would there be some **properties** about the task at hand under which the authors expect their **results to significantly change**?

The huge generalization gap of open-loop dynamics that the authors report could motivate studies in other domains. For example, LLMs are often trained through imitation learning (ie open loop), but are used in a closed loop scenario. If similar findings hold in this scenario, this would imply that a medium amount of imitation learning can actually be harmful for final performance. Overall, I feel that there are **connections to other areas of ML research**, and discussing some of them could make the paper interesting to a larger audience.

Finally, here are a few recent theory papers that the authors may find interesting and relevant to their paper. [1](https://proceedings.neurips.cc/paper_files/paper/2024/hash/fbb07254ef01868967dc891ea3fa6c13-Abstract-Conference.html) [2](https://arxiv.org/abs/2407.07279) [3](https://openreview.net/forum?id=KGOcrIWYnx) Some recent papers on learning dynamics of linear RNNs, from a more deep learning perspective than the literature the authors cite. [Here](https://journals.aps.org/prx/abstract/10.1103/PhysRevX.15.021051) is another theory paper on learning dynamics in a (simplified) RL scenario.

**Ethical Concerns:**

["NO or VERY MINOR ethics concerns only"]

**Final Justification:**

Solid paper. The authors addressed my (minor) concerns. I keep my score as it is (5).

**Limitations:**

Yes

**Quality:**

4

**Strengths And Weaknesses:**

Overall, the paper is **insightful** and **well-written**. It is **well connected** to existing work and nicely motivates why the closed-loop setting is worthwhile studying. The theoretical analysis of the learning dynamics, while specific to a particular task and simple network, is well executed and **provides non-trivial insights on interesting learning dynamics**.

Despite the limited scope of the analysis (which is needed to make progress and thus totally fine for me), I find the main weakness of the paper to be that it is **overly technical**. The big picture and the technical details are well done, but an in-between level which summarizesinterprets/justifies their results in a more intuitive way.

---

> ### Author Rebuttal · Authors · 2025-07-29
>
> Thank you very much for taking the time to carefully review our work. We appreciate your positive assessment. Your comment regarding the non-trivial insights into learning dynamics precisely captures the message we aimed to convey. Please find below our point-by-point responses to your comments and questions.
>
> **R: From my understanding, the authors use a network trained on the task as the teacher for the open-loop setting. Why not training it on the optimal control policy (which should be tractable). Given that trained networks do not reach 0 loss, their predictions might contain some strange artifacts that affect learning dynamics. Can the authors comment on that and perform, if possible, a quick experiment to verify that?**
>
> **A**: Yes, that’s correct—the teacher in our analysis is an optimized RNN. However, our results also hold for optimal control teachers. We were initially motivated by the fact that these control problems admit an analytically optimal teacher via the Riccati equation (i.e., LQR; Appendix B.1). We used an RNN teacher to ensure a fairer comparison between open- and closed-loop RNNs. Following the reviewer’s suggestion, we **performed additional experiments** using an LQR teacher and found that the sharp loss peak in the open-loop setting still persists, confirming that our results extend to this setting as well.
>
> **R: Can the authors provide intuition on l. 250 and on why low frequencies are learned faster?**
>
> **A**: The phenomenon reflects a well-documented inductive bias of neural networks to learn simpler structures in the training data before acquiring more complex components. This has been shown in various settings, including networks learning low-order moment statistics [Refinetti et al., 2023], dominant singular modes [Saxe et al., 2013], and lower frequencies before higher [Rahaman et al., 2019] (all cited in the original submission, line 265).
>
> In our case, we show that the RNN acquires a new frequency precisely when a complex conjugate outlier emerges in the spectrum of the recurrent weights (i.e., $W$), whose imaginary part matches the learned frequency (Fig. 6d, Fig. 17; original submission). Under rich learning (i.e., small initialization), outlier eigenvalues first reach lower frequencies (smaller imaginary parts) before progressing to higher ones.
>
> **R: Could the analysis provided by the authors be extended to higher-dimensional systems? Additionally, would there be some properties about the task at hand under which the authors expect their results to significantly change?**
>
> **A**: Yes, our analysis can be extended to more complex settings. Below we describe **two new experiments** that show our results hold in more complex scenarios:
>
> 1. We conducted preliminary experiments with GRU-based RNNs in a customized, higher-dimensional MiniGrid environment featuring _partial observability_. The agent starts in an ambiguous state and must localize itself to reach the goal via the shortest path. Closed-loop RNN (trained via online policy gradient) produced near-optimal behavior, whereas open-loop imitation RNN diverged and remained suboptimal, failing to develop a rich internal state. While preliminary, these results align with our theoretical findings and suggest broader applicability to nonlinear, high-dimensional tasks.
>
> 2. In response to **Reviewer 4A6e**, we systematically generalized our setup by introducing the _k_-integrator family - a cascade of integrators where the matrices $\mathbf{A}, \mathbf{B}, \mathbf{C}$ take more general forms, and the state, actuator, and observation dimensions scale to higher values (for details, see Q1 response to Reviewer 4A6e). Briefly, across **36 additional configurations**, we found that under _partial observability_, the main results of the paper hold: closed-loop RNNs exhibit a loss plateau, and open-loop RNNs show a sharp loss peak. Under _full-state feedback_, these effects largely disappear.
>
> In summary, we believe that _partial observability_ is a critical property underlying the divergence between closed- and open-loop RNNs.
>
> **R: The huge generalization gap of open-loop dynamics that the authors report could motivate studies in other domains. For example, LLMs are often trained through imitation learning (ie open loop), but are used in a closed loop scenario. If similar findings hold in this scenario, this would imply that a medium amount of imitation learning can actually be harmful for final performance. Overall, I feel that there are connections to other areas of ML research, and discussing some of them could make the paper interesting to a larger audience.**
>
> **A**: We find this direction very intriguing and are excited by the possibility that our findings could have broader implications, including for domains like LLM training. Our original motivation was to gain analytical traction on the underexplored problem of closed-loop learning in RNNs - potentially relevant to both learning theory and theoretical neuroscience. More broadly, the open- vs. closed-loop distinction is indeed relevant in other domains. For example, in the robotics and RL literature, works on imitation learning and behavioral cloning have acknowledged the pitfalls of ignoring feedback during training (Haan et al., 2019). While LLMs are beyond the scope of this work and our expertise, we appreciate the insightful suggestion and agree that exploring such connections could be a valuable direction for future research.
>
> **R: Finally, here are a few recent theory papers that the authors may find interesting and relevant to their paper. [1] , [2] , [3] Some recent papers on learning dynamics of linear RNNs, from a more deep learning perspective than the literature the authors cite. [Here] is another theory paper on learning dynamics in a (simplified) RL scenario.**
>
> **A**: We thank the reviewer for pointing out these interesting and relevant works. Some are already cited in the original submission - for example, [3] and the RL perceptron paper [Here], which was one of the early motivations for this project. We will carefully review the remaining suggestions and make sure to integrate them into the revised manuscript.

---

### Official Review · Reviewer_iiBW · 2025-07-01

**Clarity:** 3
**Significance:** 4
**Originality:** 4
**Rating:** 5
**Confidence:** 4

**Summary:**

This paper analyzes recurrent neural networks (RNNs) trained to control a dynamical system in a closed-loop environment where the RNN dynamics influence the external state dynamics. The authors show that the learning trajectories of RNNs trained with gradient descent differ substantially between open-loop and closed loop environments, with open loop training leading to large overfitting peaks. The authors provide simulations in nonlinear RNNs and a three-stage analysis of the learning dynamics for linear RNNs trained in closed-loop on a double integrator task $\ddot x(t) = u(t)$. In the first stage, the RNN learns a "negative position" controller where $u(t) \approx - x(t)$. In the second stage, the model begins to infer velocity after a plateau in the performance. In the final stage, the model refines the scale of the readout weights, leading to convergence. The authors also describe multi-frequency tasks in the closed loop setting.

**Questions:**

1. Throughout the paper, the authors often try to compare gradient descent dynamics on the parameters, possibly implicitly assuming that they should coincide to gradient descent with respect to the order parameters (such as $\sigma$ or the eigenvalues of $P$, etc). I am skeptical of this claim. For example, a gradient with respect to weights $W$ will not coincide with a gradient with respect to an eigenvalue of $P$. Could the authors provide some clarification on this point?
2. I did not understand the motivation to study $L = \alpha L(t) + (1-\alpha) L(0)$. What value of $\alpha$ should correspond to the actual experiments?
3. In the multi-frequency task, what sets the timescale of learning each distinct frequency? In general, if the model requires generating $m$ distinct frequencies, do the dynamics close in a $m$ dimensional subspace?

**Ethical Concerns:**

["NO or VERY MINOR ethics concerns only"]

**Limitations:**

The authors do a good job of discussing the limitations of their work in the discussion.

**Quality:**

3

**Strengths And Weaknesses:**

Strengths:

1. This paper studies an important problem (how neural networks learn to control an agent that interacts with its environment), which has until now evaded theoretical study.
2. The paper has an in depth analysis of the learning dynamics in three distinct stages and supports their results with simulations.
3. The authors also extend the analysis to multi-frequency tasks.
4. There are many other interesting results in their Appendix which provides comparisons between SGD and Adam, comparisons to optimal control, etc.

Weaknesses:

1. To make theoretical analysis tractable, the authors focus their theoretical efforts on a rather double integrator system.
2. A few of the mathematical / modeling assumptions seem slightly dubious (see questions), but I am happy to be convinced if I am wrong or misunderstanding.

Overall, this is a novel paper studying an important problem in the theory of RNNs and I advocate acceptance.

---

> ### Author Rebuttal · Authors · 2025-07-29
>
> We thank the reviewer for the positive feedback and for recognizing both the novelty and importance of the problem addressed. We also appreciate your support for the acceptance of our work. Please find below our point-by-point responses to your questions.
>
> **R: Throughout the paper, the authors often try to compare gradient descent dynamics on the parameters, possibly implicitly assuming that they should coincide to gradient descent with respect to the order parameters (such as $\sigma$ or the eigenvalues of $P$, etc). I am skeptical of this claim. For example, a gradient with respect to weights $W$ will not coincide with a gradient with respect to an eigenvalue of $P$. Could the authors provide some clarification on this point?**
>
> **A**: We fully agree with the reviewer's keen observation that gradients w.r.t. the full RNN weights ($\boldsymbol{W}$) do not, in general, coincide with gradients taken w.r.t. the low-dimensional order parameters such as $\sigma$ or the eigenvalues of $P$. This point was noted in the original submission (lines 159–160):
>
> *"Gradients applied to the order parameters in $P_{\text{eff}}$ do not directly correspond to weight updates in the full RNN".*
>
> Our goal, rather, was to show that the order parameters serve as a compact and interpretable proxy for understanding learning dynamics, as they clearly capture the three learning phases observed in the full model. It turns out that despite this mismatch, the loss dynamics in Stage 3 are almost identical when using the order parameters, and in Stage 2, they differ only in the timing of the plateau.
>
> Furthermore, since the original submission, we have begun to develop exact closed-form expressions for the weight updates in terms of the order parameters alone. Preliminary results indicate that this allows an exact match to the high-D dynamics and has implications beyond the closed-loop setting. We believe that adding these results would make an already large appendix even larger and would neither allow for a full discussion of these implications, nor for a proper review. We are, however, open to different views on this matter.
>
> **R: I did not understand the motivation to study $L = \alpha L(t) + (1-\alpha) L(0)$. What value of $\alpha$ matches the actual experiments?**
>
> **A**: We introduced the blended loss as a mathematically tractable surrogate that approximates the full loss and provides intuition about the competing objectives the RNN faces in Stage 2. The full loss averages over all timesteps from $0$ to $T$, but it is analytically unwieldy and not easily readable. Our surrogate replaces it with two interpretable components, expressed in terms of the order parameters:
>
> 1.  Short-term control - minimizing initial position error, captured by $\mathcal{L}_0$, which has a simple closed-form (Eq. 13).
>
> 2.  Long-term stability - pushing the dominant eigenvalues of $P$ inside the unit disk, captured by $\mathcal{L}_\infty$ (Eq. 11), based on the norm of the leading eigenvalue. Although this can be computed exactly at the onset of Stage 2, it later becomes the root of a cubic polynomial and offers limited additional insight.
>
> Optimising either term in isolation does not reproduce the empirical eigenvalue trajectory (Figs. 4 c-e):
> 1. $\alpha = 0$ (short‑term only) - drives the dominant eigenvalue upward along the imaginary axis.
> 2. $\alpha = 1$ (long‑term only) - yields a direct vertical descent. Note that gradient descent on the order parameters causes only the imaginary part to decrease, and not towards the origin.
>
> Blending the two $0 < \alpha < 1$ produces inward trajectory (Fig. 4d), validating $\mathcal{L}_{\text{eff}}$ as a faithful low‑dimensional surrogate.
>
> **A:** As for your question regarding the value of $\alpha$ - In the experiments, the RNN is trained on finite episode lengths $T$, and $\alpha$ can be thought of as a knob that replicates the effect of training with various $T$. In Fig. 4e, we validate this theoretical prediction by training identical initial‑condition RNNs with various episode lengths $T$ and showing that they qualitatively produce the trajectories predicted by our surrogate loss using $\alpha$ (Fig. 4d).
>
> We do not claim to derive an exact mapping between $\alpha$ and $T$ - the dominant eigenvalue trajectory could depend on other factors as well. We merely claim that the chosen $\alpha$ in our surrogate loss ($\mathcal{L}_{\text{eff}}$) provides a good qualitative fit to the phenomenon observed during Stage 2 in the high‑D model.
>
> Furthermore, motivated by your question, we **ran new numerical experiments** sweeping over a finer grid of surrogate parameters $\alpha$ and episode lengths $T$. For each, we computed the phase of the dominant complex eigenvalue. The resulting scatter plot traces a smooth curve, revealing a clear $\alpha - T$ correlation that provides additional validation for our surrogate-loss intuition.
>
> **R: In the multi-frequency task, what sets the timescale of learning each distinct frequency?
> In general, if the model requires generating $m$ distinct frequencies, do the dynamics close in a $m$  dimensional subspace?**
>
> **A**:  Regarding your first question about the timescale of learning each frequency, we find this phenomenon particularly intriguing. Learning appears to unfold in discrete, well-delineated stages, marked by sharp drops in training loss. Empirically, the end of each plateau aligns precisely with the appearance of a new complex-conjugate eigenvalue pair in the recurrent matrix $\boldsymbol{W}$ (Fig. 6d, Fig. 17), whose imaginary part matches the newly acquired frequency. In the multi-frequency task, we did not attempt to derive exact analytical timescales, as this section aimed primarily to test the generality of the double-integrator findings, using fewer modeling constraints (e.g., Adam optimizer, full-rank weights).
>
> For the double-integrator task, we attempted a more analytical approach. Our analysis suggests that the single plateau ends in a bifurcation that stabilizes the coupled system, but solving for the precise timescales requires handling the eigenstructure of a cubic polynomial—analytically cumbersome even with symbolic math. We believe this is an important and promising direction for future work.
>
> **A**: Regarding your second question, whether dynamics close in an $m$ dimensional subspace. We ran a **new analysis** to test this. Specifically, we pooled hidden states from 100 evaluation rollouts and performed PCA at five checkpoints: epoch 0 and the four loss drops shown in Fig. 6b. The results revealed a clear pattern: each loss drop introduces a new principal component with substantial variance. At epoch 0, the first PC alone explains ~80% of the variance, while by the final loss drop, the top $m$ PCs are needed to explain >90% of the variance.

---

> > ### Comment · Reviewer_iiBW · 2025-08-04
> >
> > I thank the reviewers for their detailed responses to my questions and also for their efforts in testing out some unknowns with **new experiments** (I appreciate that this takes effort). I now better understand the motivation for the blended loss and am intrigued by the new PCA experiment in the multi-frequency task.
> >
> > Overall, I will maintain my positive score and vote of acceptance for this paper.

---

> > > ### Author Response · Authors · 2025-08-05
> > >
> > > Thank you very much for your response and support for accepting our work. We're pleased our reply helped clarify the motivation behind the blended loss (we’ll make sure to reflect this in the revised manuscript) and that the new experiments were of interest (we share your enthusiasm for this direction).

---

### Official Review · Reviewer_4A6e · 2025-07-02

**Clarity:** 3
**Significance:** 3
**Originality:** 3
**Rating:** 4
**Confidence:** 4

**Summary:**

In this paper, the authors analyze the learning dynamics of a closed-loop RNN performing a double integrator control task. They (mostly) work in the setting of a linear RNN, and perform their analysis based on the spectrum of a simplified version of the resulting coupled linear system. They identify unique dynamics emanating from the coupling with the environment, and distinguish these dynamics from an open-loop model that is trained in a student-teacher setup.

**Questions:**

1. In equation (1), the authors chose a very specific problem. How much of their analytic analysis can be extended to a setup with somewhat more general matrices $\mathbf{A}$, $\mathbf{B}$, $\mathbf{C}$ ?
2. On line 70, pg. 2, the authors assume that $N$ is large. Where is that assumption used?
3. In fig. 1 (c), why is the initial train loss lower than the initial test loss?
4. In line 98, when the authors state that they use SGD for training, do they mean BPTT?
5. How is the gradient clipping taken into account for the analytical analysis?
6. In fig. 2 (a), can the authors characterize the resonance in the open loop training, and understand the transition to stability?
7. In line 165 on page 2 on page 5, can the authors support the statement that $u_t \propto -\text{pos.}$? Fig. 3 (b) by itself doesn't seem sufficient to support this.
8. On line 168, how do the authors support the claim that the velocity is not represented in the RNN?
9. In equation (14), the surrogate loss seems ad hoc. If the authors claim that the gradient induced by this loss is correlated to the actual gradient, can they support this?

**Ethical Concerns:**

["NO or VERY MINOR ethics concerns only"]

**Final Justification:**

The authors study a problem of considerable relevance to the field of learning dynamics. While their analysis relies on surrogate losses and approximating gradient descent dynamics through order parameters, their results are intuitive and interesting. I (narrowly) recommend acceptance, given the more general analysis the authors have performed in the meantime.

**Limitations:**

See above.

**Quality:**

2

**Strengths And Weaknesses:**

## Strengths

The authors study an important problem, and provide good intuition for the different training stages. I didn't find any major flaws in their analysis, and I am convinced this project has great potential.

## Major Weakness
1. Studying the setup for a linear RNN, especially in the continuous-time description, seems to provide the opportunity for very explicit analyses, for example, based on the explicit loss functions and gradients, generalizations to a whole class of control problems and the effects of regularization. While the authors provide valuable insights into the specific setups they chose, it seems that their analysis is unfinished.

## Minor Weaknesses
2. Fig. 2 (b) and (c) are hard to read and understand, because of the contrast, the size and the unexplained arrows.
3. In line 145 on page 5, the statement about generality seems hard to justify.

---

> ### Author Rebuttal · Authors · 2025-07-29
>
> We thank the reviewer for appreciating the importance of our work.
>
> **R: Studying the setup for a linear RNN, especially in the continuous-time description, seems to provide the opportunity for very explicit analyses, for example, based on the explicit loss functions and gradients, generalizations to a whole class of control problems and the effects of regularization. While the authors provide valuable insights into the specific setups they chose, it seems that their analysis is unfinished.**
>
> **A**: We appreciate the reviewer’s acknowledgment of the insights we provide, but we believe some aspects of the critique may stem from misunderstandings, which we are happy to clarify.
>
> 1. **"continuous-time description"** - Neither our main text nor theoretical analysis involves a continuous-time model. Both the RNN and environment are defined in **discrete time** (lines 71–72: _“We adopt a discrete‑time RNN to align with the discrete nature of the control task.”_). The Appendix (A.1) contains continuous time results as a robustness check and is not central to our analysis.
>
> 2.  **"based on the explicit loss functions and gradients"** – The loss function is not given explicitly, as it depends on solving the eigenvalues and eigenvectors of a 4D matrix (or equivalently, unfolding the RNN and environment dynamics over $T$ steps). This is precisely why we used a surrogate loss (see also our response to **Reviewer iiBW**).
>
> 3.  **"generalization to broader control problems"** - See our response to your **Q1** below (k-integrator) and to **Reviewer xBz7** (MiniGrid).
>
> 4. **"effects of regularization"** - We have a dedicated Appendix (section A.2) on this topic in the original submission.
>
> **R: Fig. 2 (b) and (c) are hard to read and understand, because of the contrast, the size and the unexplained arrows.**
>
> **A**: We will improve the contrast and size in the final version. As for the arrows, we are unsure which unexplained arrows the reviewer is referring to. **Fig 2b** includes arrows linking "Start" to "End" to indicate the direction of training over epochs. **Fig. 2c** shows the same arrows as in Fig. 2a. The caption also explains the relationship between the gray and yellow markers across panels. If the reviewer has more specific concerns about figure clarity, we would be happy to address them.
>
> **R: In line 145 on page 5, the statement about generality seems hard to justify.**
>
> **A**: In line 145, we write: _“these assumptions simplify analysis without sacrificing generality”_, referring to our use of a rank-1 linear RNN with fixed input weights to enable analytical treatment. Crucially, this simplified model still exhibits the same 3-stage learning dynamics as the full-rank nonlinear RNN with all weights trainable. Thus, the core phenomena are preserved, and the claim of generality is supported by our results. We will clarify this more explicitly in the revision.
>
> **R: 1. In equation (1), the authors chose a very specific problem. How much of their analytic analysis can be extended to a setup with somewhat more general matrices ${\mathbf{A}, \mathbf{B}, \mathbf{C}}$?**
>
>  **A**: We agree that our starting point is a specific problem. We chose the double-integrator because it is a well-established benchmark in control theory (e.g., Recht, 2019; Friedrich et al., 2021), which allowed us to make concrete progress on the underexplored question of how RNNs learn in closed-loop settings.
>
> To our knowledge, one relevant prior work is Razin et al., 2024 (cited on line 279), which considers arbitrary $\mathbf{A}$ and $\mathbf{B}$ but assumes **full-state feedback**. While other work may exist, this study motivated our focus on the **partially observable** regime ($\operatorname{rank}(C) < \operatorname{rank}(A)$; Eq.2, line 65), which is also more aligned with biologically inspired settings. Notably, even in the minimal double-integrator task, this setup reveals a clear divergence between closed- and open-loop learning - the central phenomenon our paper aims to explain.
>
> As the reviewer requested, we ran **additional extensive experiments** to test the generality of our findings by introducing the _k_-integrator family - a cascade of integrators where $\mathbf{A}, \mathbf{B}, \mathbf{C}$ take more general forms, spanning both partially observable and full-state feedback cases. In total, we evaluated **36 additional configurations** beyond the one in the main text, varying the state dimension ($k \in {2, 3, 4, 5}$), number of actuators ($b$), and number of observed variables ($c$). Key results are summarized below:
>
> - Partial observability ($c < k$): we recover both key phenomena reported in the original submission- a loss plateau in closed-loop RNNs and a sharp loss peak in open-loop RNNs.
>
> - Full-state feedback ($c = k$): As expected, the loss curves of closed- and open-loop RNNs are quite similar, with much less pronounced plateaus or sharp peaks.
>
> **R: 2. On line 70, pg. 2, the authors assume that $N$ is large. Where is that assumption used?**
>
> **A**: This is the common scenario in theoretical RNN/neuroscience studies, and hence we used it. The assumption that $N$ is large is not critical to our analysis. We used $N = 100$ (line 94), but also tested $N = 50$, $200$, and $500$, and found the learning dynamics to be invariant to $N$.
>
> **R: 3. In fig. 1c, why is the initial train loss lower than the initial test?**
>
> **A**: The discrepancy arises because we mistakenly recorded the **test loss at epoch 0** _before_ any updates, while the **train loss at epoch 0** was evaluated _after_ the first update. Thanks for pointing this out, and we will fix this in the final version.
>
> **R: 4. In line 98, when the authors state that they use SGD for training, do they mean BPTT?**
>
> **A:** Backpropagation through time is a method to compute gradients. Stochastic gradient descent is a method to choose samples on which the gradient is computed. These refer to different parts of the training process. We use BPTT as a way to compute gradients for SGD. If the reviewer had a different concern in mind, we’d be happy to clarify further.
>
> **R: 5. How is the gradient clipping taken into account for the analytical analysis?**
>
> **A**: Gradient clipping is used in practice to prevent numerical overflow from unstable closed-loop policies - a simulation artifact, not a meaningful physical effect. We can also avoid clipping by clamping the environment (i.e., restricting position or velocity from becoming too large). In the multi-frequency task (Section 5), where we test the generality of our findings, no gradient clipping was applied - yet both key phenomena (loss plateau in closed-loop, sharp loss peak in open-loop) still appear clearly (Fig. 6b). In Figs. 4 and 5, we applied gradient clipping to the order parameters for the same reason.
>
> **R: 6. In fig. 2 (a), can the authors characterize the resonance in the open loop training, and understand the transition to stability?**
>
> **A**: The term **resonance** is not used anywhere in the paper, nor is it conceptually relevant to our analysis, so it's unclear how to respond to this specific point. **Fig. 2a** shows a transient peak in error over time, which may bear some visual resemblance to resonance curves. The transition to stability, however, is thoroughly described as Stage 2 of learning (Section 4.2), with dedicated illustrations in Figs. 2b and 4.
>
> **R: 7. In line 165 on page 2 on page 5, can the authors support the statement that $u(t) \propto -\text{pos}$.? Fig. 3 (b) by itself doesn't seem sufficient to support this.**
>
> **A**: Yes, the statement is supported both numerically and analytically in the paper:
>
> 1.  **Numerically**: In Figs. 2 b,c, during Stage 1, the closed-loop RNN converges to effective linear gains with $k^{\text{eff}}_1 < 0$ and $k^{\text{eff}}_2 \approx 0$, which corresponds to a policy of the form $u_t \propto -\text{pos}$.
>
> 2. **Analytically**: In Section 4.2 (Stage 1), we assume small initial recurrent weights ($W \approx 0$), so the output is dominated by the input–output overlap $\sigma_{zm}$. With $\sigma_{zm} < 0$, the RNN implements $u_t \propto -\text{pos}$.
>
> 3.  **Validation**: Our analysis shows that in Stage 1, the loss depends only on $\sigma_{zm}$, enabling an exact loss derivation (Fig. 3d). This is later confirmed in simulations: RNNs with different initial $\sigma_{zm}$ values all converge to $\sigma_{zm} < 0$ by the end of Stage 1 (Fig. 3e).
>
> **R: 8. On line 168, how do the authors support the claim that the velocity is not represented in the RNN?**
>
> **A**: This is addressed in Section 4.2 (Stage 2). Briefly, the RNN receives **only position** as input - **velocity is never observed** (lines 64–66). Early in training, with negligible recurrent weights ($W \approx 0$), the effective gain on velocity is near zero ($k_2^{\text{eff}} \approx 0$), and the policy reduces to $u_t \propto -\text{pos}$ (see response to Q7 above). Since the RNN lacks both velocity input and internal memory early on, it cannot represent velocity. Only later, as $W$ increases, does the network begin to infer velocity by integrating across time (e.g., via $x_t^{(1)} - x_{t-1}^{(1)}$).
>
> **R: 9. In equation (14), the surrogate loss seems ad hoc. If the authors claim that the gradient induced by this loss is correlated to the actual gradient, can they support this?**
>
> **A**: We agree that the surrogate loss in Eq. (14) is ad hoc. Its value lies in its tractability and ability to provide intuition: it depends only on order parameters (e.g., the $\sigma$’s) and captures the key tradeoff in closed-loop learning - short-term control ($\mathcal{L_0}$) vs. long-term stability ($\mathcal{L_\infty}$) - modulated by a single tuning parameter $\alpha$. We also refer the reviewer to our response on the surrogate loss provided to **Reviewer iiBW**.

---

> > ### Comment · Reviewer_4A6e · 2025-08-05
> >
> > I want to thank the authors for directly addressing my main concerns. While i believe that the analysis is not yet 100% complete, I understand the authors' concerns about extending the scope further. I'll raise my score, assuming that the integrator generalization will appear in the camera-ready version of the article, and I'd encourage the authors to continue their studies in a follow-up paper.

---

> > > ### Author Response · Authors · 2025-08-05
> > >
> > > Thank you very much for your response and the increase in score. We’re pleased that our new experiment addressed your main concern and appreciate that your comment motivated us to carry it out. We will make sure to include this in the revised manuscript.

---

### Author Response · Authors · 2025-08-06
**General Response**

We thank the reviewers for their detailed feedback, valuable ideas, and overall positive response. We’re glad that the new results addressed key concerns and helped strengthen our submission. We also appreciate the reviewers’ engagement during the rebuttal phase.

We plan to incorporate two main additions in the revision, directly addressing points raised by the reviewers:

1. New experiments testing the generality of our findings, including alternative LQR teachers and generalization to a broader class of control problems (i.e., _k_-integrator).

2. Improved clarity and narrative flow to make the paper more accessible, including a clearer motivation for the surrogate loss, the role of optimization over order parameters, and intuition for when these approximations are effective or fail.

We are thankful that reviewers *iiBW* and *xBz7* support the acceptance of our work, describing it as novel (*iiBW*), addressing an important and under-studied problem (*iiBW*), with in-depth (*iiBW*) and well-executed (*xBz7*) theoretical analysis, and presenting non-trivial and insightful results (*xBz7*). We also thank *4A6e* for recognizing the importance of our work and motivating the generalization experiments that help strengthen our paper.

---

### Decision · Program_Chairs · 2025-09-17

**Decision:**

Accept (poster)

**Comment:**

The paper studies training RNNs in open-loop (supervised) or closed-loop (influencing it's own next input) setting, performing a double integrator control task. First, the paper studies that the same RNN, under these two settings, undergoes drastically different learning stages.

Strengths: The paper is clear and well-written and provides non-trivial insights into these understudied learning dynamics.
Weaknesses: Some reviewers would have welcomed more empirical investigations outside the mathematical tractable setting this works studies. I find this not necessary.

The paper is clearly well-executed and, after the rebuttal, no (major) weaknesses remain.